eLIFE elifesciences.org

# Selective targeting of unipolar brush cell subtypes by cerebellar mossy fibers

**Timothy S Balmer, Laurence O Trussell***

Vollum Institute and Oregon Hearing Research Center, Oregon Health and Science University, Portland, United States

**Abstract** In vestibular cerebellum, primary afferents carry signals from single vestibular end organs, whereas secondary afferents from vestibular nucleus carry integrated signals. Selective targeting of distinct mossy fibers determines how the cerebellum processes vestibular signals. We focused on vestibular projections to ON and OFF classes of unipolar brush cells (UBCs), which transform single mossy fiber signals into long-lasting excitation or inhibition respectively, and impact the activity of ensembles of granule cells. To determine whether these contacts are indeed selective, connectivity was traced back from UBC to specific ganglion cell, hair cell and vestibular organ subtypes in mice. We show that a specialized subset of primary afferents contacts ON UBCs, but not OFF UBCs, while secondary afferents contact both subtypes. Striking anatomical differences were observed between primary and secondary afferents, their synapses, and the UBCs they contact. Thus, each class of UBC functions to transform specific signals through distinct anatomical pathways.

DOI: https://doi.org/10.7554/eLife.44964.001

## Introduction

In the cerebellum, mossy fibers convey multimodal signals from diverse regions of the central nervous system to the granule cell layer. 'Expansion recoding' theories of cerebellar processing require these mossy fiber inputs to diverge to hundreds of granule cells, and their signals are integrated first by granule cell dendrites and then by the massive dendritic trees of Purkinje cells (***Albus, 1971***). However, mossy fiber inputs to vestibular cerebellum differ from those of other cerebellar lobes in receiving mossy fibers directly from peripheral ganglion neurons (the primary vestibular afferents), as well as from brainstem nuclei (secondary afferents). Each of the primary afferents carries signals from a single vestibular organ, with each organ coding head position or velocity in a given plane (***Fernández et al., 1988***). Maintaining separate mossy fiber signals from specific end organ sources in 'labeled lines' could allow segregated ensembles of granule cells to faithfully encode head movements along specific planes. Alternatively or additionally, specific sources might undergo selective amplification by local circuitry to enhance their representation to downstream neurons. However, such mechanisms are not consistent with the general view of cerebellar function that diverse mossy fibers are simply integrated by individual granule cells, and differ primarily by short term plasticity at the granule cell synapse (***Chabrol et al., 2015***; ***Huang et al., 2013***), necessitating an examination of mossy fiber innervation of the vestibular cerebellum.

We investigated this problem by tracing vestibular cerebellar mossy fibers back to their primary and secondary sources, and forward to target neurons in the cerebellum, focusing on unipolar brush cells (UBCs), because they form a processing layer prior to the well-studied granule cells. UBCs are excitatory interneurons within the granule cell layer that receive a single mossy fiber ending on their brush-like dendrite (***Harris et al., 1993***; ***Mugnaini and Floris, 1994***; ***Mugnaini et al., 2011***). Instead of integrating multiple inputs as granule cells do, this large synaptic contact dramatically transforms activity of one mossy fiber before projecting to hundreds of granule cells and other UBCs. In

***For correspondence:** trussell@ohsu.edu

**Competing interests:** The authors declare that no competing interests exist.




**eLife digest** While out jogging, you have no trouble keeping your eyes fixed on objects in the distance even though your head and eyes are moving with every step. Humans owe this stability of the visual world partly to a region of the brain called the vestibular cerebellum. From its position underneath the rest of the brain, the vestibular cerebellum detects head motion and then triggers compensatory movements to stabilize the head, body and eyes.

The vestibular cerebellum receives sensory input from the body via direct and indirect routes. The direct input comes from five structures within the inner ear, each of which detects movement of the head in one particular direction. The indirect input travels to the cerebellum via the brainstem, which connects the brain with the spinal cord. The indirect input contains information on head movements in multiple directions combined with input from other senses such as vision.

By studying the mouse brain, Balmer and Trussell have now mapped the direct and indirect circuits that carry sensory information to the vestibular cerebellum. Both types of input activate cells within the vestibular cerebellum called unipolar brush cells (UBCs). There are two types of UBCs: ON and OFF. Direct sensory input from the inner ear activates only ON UBCs. These cells respond to the arrival of sensory input by increasing their activity. Indirect input from the brainstem activates both ON UBCs and OFF UBCs. The latter respond to the input by decreasing their activity.

The vestibular cerebellum thus processes direct and indirect inputs via segregated pathways containing different types of UBCs. The next step in understanding how the cerebellum maintains a stable visual world is to identify the circuitry beyond the UBCs. Understanding these circuits will ultimately provide insights into balance disorders, such as vertigo.

DOI: https://doi.org/10.7554/eLife.44964.002

vestibular processing areas of rodent cerebellum, UBCs are present in exceptionally high density and could coordinate ensembles of granule cells to respond to single directions of movement (**Floris et al., 1994**).

This problem is deepened by the diversity of UBCs. Two subtypes of UBC have been described: ON UBCs respond to mossy fiber input with a prolonged depolarization and enhancement of firing while OFF UBCs are inhibited (**Borges-Merjane and Trussell, 2015**). Both responses last for hundreds of milliseconds, an outcome of selective receptor expression in the two subtypes (**Borges-Merjane and Trussell, 2015**) combined with the great size the mossy fiber-UBC synaptic contact (**Mugnaini et al., 1994**). Given this potent circuit element, it is critical to determine which vestibular organs map directly to cerebellum and which UBC subtypes they contact to understand vestibular representation. For example, if both subtypes receive common sensory input, then the ON/OFF distinction in UBCs would allow mossy signals to diverge, setting up distinct processing pathways within the granule cell layer, such that the OFF pathway could be a negative image of the vestibular motion. On the other hand, if each subtype receives mossy fiber input that conveys a distinct vestibular modality, then ON and OFF UBCs would mediate modality-specific transformations of extrinsic inputs. Here we show that in cerebellar lobe X of mouse, the primary representation is from a subset of angular acceleration coding neurons, and these signals reach and are amplified by ON UBCs, but not OFF UBCs. OFF UBCs by contrast only process secondary afferent signals that may contain signals integrated over multiple directions of movement, hemispheres and/or modalities. We also show that primary and secondary inputs exhibit dramatic differences in their axonal and synaptic morphology, as well as in the morphology of the UBCs they contact, which may further refine coding in the granule cell layer.

# Results

## Primary vestibular afferents in the Glt25d2 mouse line

Vestibular hair cells detect head acceleration, velocity, and gravity, and convey these signals to vestibular ganglion (VG) neurons; within each vestibular end organ, there are two subtypes of hair cell and at least three subtypes of VG cell (**Eatock and Songer, 2011**). VG axons project to vestibular nuclei in the brainstem and directly to the ipsilateral vestibular cerebellum (**Dow, 1936**). First, we set

out to determine which VG cells project centrally to cerebellum and, peripherally, which end organ and hair cell subtype those same neurons contact. In order to express transgenes in primary afferents that may project to UBCs in the vestibular cerebellum, we determined that the Glt25d2 mouse line that has Cre recombinase (Cre) targeted to the *Colgalt2* locus (B6-Tg(Colgalt2-cre)NF107Gsat) (**Gerfen et al., 2013**) expresses Cre in primary afferents projecting to the vestibular nuclei and vestibular lobes of the cerebellum, by crossing it with a tdTomato reporter line (Ai9) (**Figure 1A–B**). In cerebellar lobes IX and X, these afferents appeared as mossy fibers, and were most likely primary (first-order) from the VG, and not those from brainstem vestibular nuclei or nucleus prepositus hypoglossi that also project to cerebellum, because no somata lying in these areas expressed Cre (**Figure 1C**). Primary afferents did not project to flocculus or paraflocculus.

VG neurons have specialized dendrites that receive input from vestibular hair cells in the five vestibular end organs: the three semicircular canals and the two otolith organs, the utricle and sacculus. There are 3 types of peripheral afferent neuron based on their dendritic morphology: 'pure-calyx', which form calyx endings on Type I hair cells, bouton, which makes bouton endings on Type II hair cells, and dimorphic, which have both calyx and bouton terminals (**Fernández et al., 1988**). The central regions of each end-organ are populated with 'pure-calyx' type dendritic endings of VG neurons expressing calretinin (**Desmadryl and Dechesne, 1992**; **Leonard and Kevetter, 2002**). Note that pure-calyx endings also receive input from Type II hair cells that contact the outer surface of the calyx. tdTomato-positive VG neurons in the Glt25d2::tdTomato mouse varied in soma size, location and calretinin expression (**Figure 1D–E**), indicating Cre expression in diverse types of VG neurons. Indeed, some peripheral afferents that expressed tdTomato had pure-calyx endings (based on co-labeling with calretinin) and others had dimorphic endings (**Figure 1F–G**). It was not possible to determine whether pure bouton endings expressed tdTomato because pure bouton endings could not be differentiated from boutons extending from the dimorphic fibers.

## Cre+ dimorphic vestibular afferents from semicircular canals project to cerebellum

To determine which signals are carried to cerebellum via Cre+ primary afferents in the Glt25d2 mouse line, we used retrogradely-infecting adeno-associated viruses (retro-AAVs) that express GFP. Unlike typical AAVs, retro-AAVs infect axons and thus allow the source of projections to the injected site to be determined (**Tervo et al., 2016**). Injections of Cre-dependent retro-AAV (AAV2-retro-CAG-Flex-GFP) were made into lobe X to label projecting VG neurons and their peripheral afferents in the five vestibular end organs (**Figure 2A**).

GFP-expressing afferents were apparent at the injection site in lobe X (**Figure 2B**). The VGs ipsilateral to the injected side of lobe X were immunolabeled for calretinin and imaged as whole mounts (**Figure 2C**). In a total of 670 retrolabeled VG somata, none expressed calretinin (n = 5 ganglia in separate experiments), suggesting that Cre-positive (Cre+) cells with pure-calyx endings do not project to lobe X. Note that we could not be confident we imaged every VG neuron because dissection of the complete VG complex was not always possible.

In two Glt25d2 mice, histological analysis of the injection site revealed numerous GFP-labeled primary afferents in lobe X and very few in lobe IX. In these experiments, all five end organs and VG ipsilateral to the injection site were successfully processed. The afferents in the end organs were almost exclusively dimorphic, having calyx endings surrounding hair cells with extending processes ending in boutons (**Figure 2D–H**). No pure-calyx endings and only one bouton-only ending was observed (**Figure 2C**). No retrolabeled afferents were co-labeled with calretinin (0 of 380 calyces, two mice, 10 end organs), consistent with the counts of labeled VG somata. Most of the retrolabeled afferents surrounded hair cells in the semicircular canals and the sacculus (anterior canal, 55, 64; horizontal canal, 22, 35; posterior vertical canal, 55, 50; utricle, 5, 6; sacculus; 53, 35; numbers are retrolabeled calyces in each experiment where injections were restricted to lobe X). As expected, only a few afferents in the end organs contralateral to the injection were retrolabeled, likely due to virus that diffused across the cerebellar midline (**Korte and Mugnaini, 1979**).

Another injection labeled numerous afferents projecting to the ventral leaflet of lobe IX in addition to lobe X. As expected, there were more afferents labeled in all end organs (anterior canal, 109; horizontal canal, 49; posterior canal, 203; utricle, 109; sacculus, 259) (**Figure 2—figure supplement 1**). Despite many more afferents labeled, none expressed calretinin and were therefore also exclusively not pure-calyx endings (0 of 729 calyces, one mouse, five end organs). The result that many

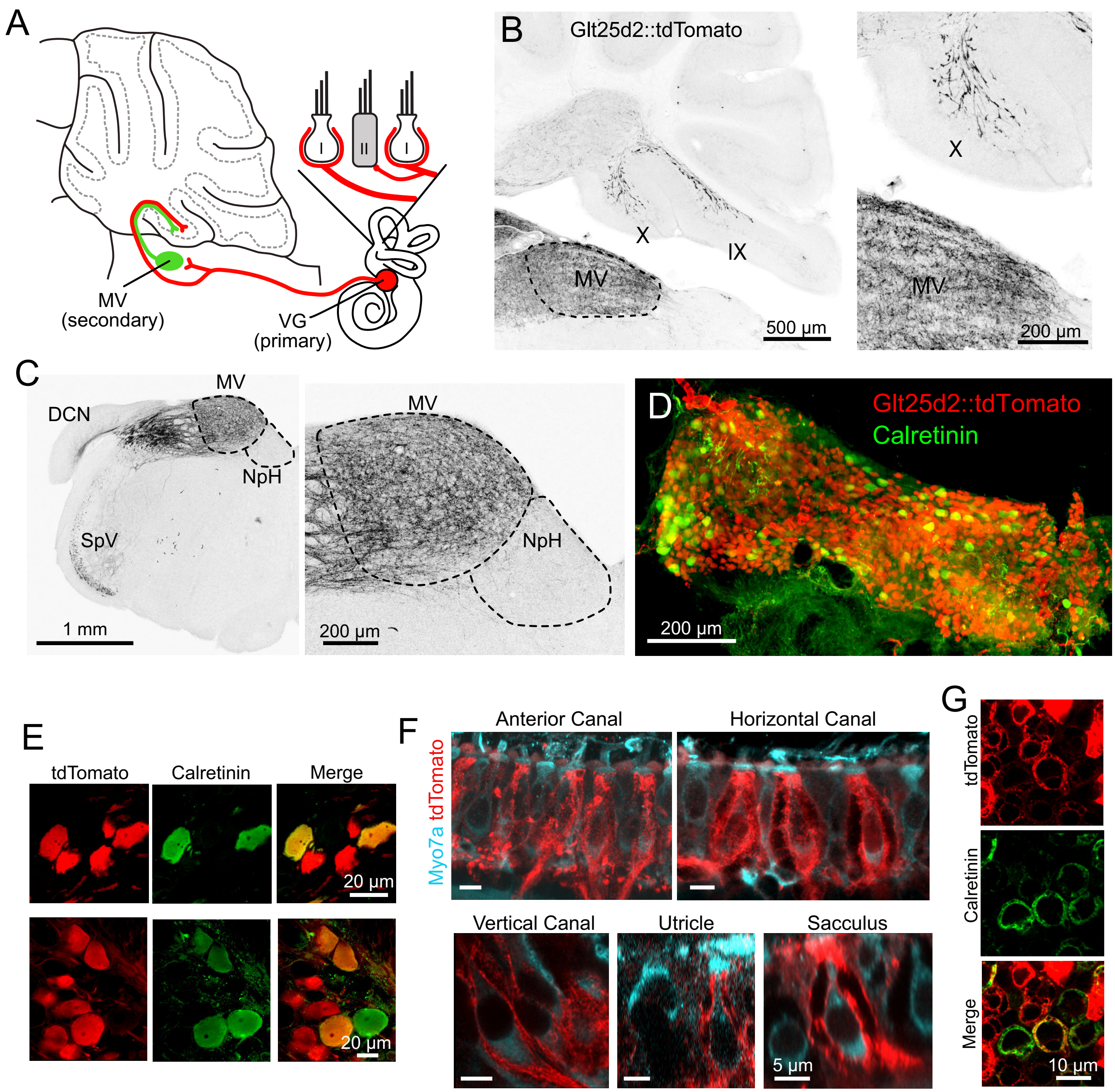


**Figure 1.** Glt25d2 mouse line expresses Cre in VG neurons, their peripheral afferents and their primary projections to vestibular nucleus and vestibular cerebellum. (**A**) Schematic of the vestibular cerebellar circuit. Primary afferent neuron somata in the VG have peripheral afferent dendrites that end in calyces, boutons, or both (dimorphic) recieving input from type I and/or type II hair cells in one of five vestibular end organs. Their axons project into the vestibular nuclei inclduing medial vestibular nucleus (MV) and into the vestibular lobes of cerebellum. MV neurons integrate input from multiple primary afferents carrying information from multiple end organs, in addition to inputs from other parts of the brain. MV neurons provide secondary mossy fiber input to the vestibular cerebellum. (**B**) Left- Sagittal section of Glt25d2::tdTomato brain showing mossy fibers projecting into lobe X and ventral lobe IX. Note that the medial vestibular nucleus (MV) is innervated by primary fibers but that the local neurons do not express tdTomato (no labeled somata), indicating that the labeled mossy fibers are not secondary vestibular afferents from this nucleus. Right- Magnification of lobe X showing tdTomato+ primary afferents. (**C**) Left- Coronal section showing tdTomato+ fibers (black). Note auditory nerve fibers in dorsal cochlear nucleus (DCN) and trigeminal nerve fibers in spinal trigeminal nucleus (SpV). Nucleus prepositus hypoglossi (NpH), which is known to project to lobe X, has no labeled somata. Right- Magnification of MV and NpH. (**D**) Whole-mount of VG showing tdTomato expression and colabeling for calretinin. (**E**) Two

*Figure 1 continued*

example areas of VG with a subset of tdTomato-expressing cells colabeled for calretinin. (F) In all five end organs, tdTomato+ peripheral afferents were found. Myo7A (cyan) was used to label hair cells. (G) Example of calyces of the utricle (view from the top) that express tdTomato and in some cases colabeled for calretinin, indicating their pure-calyx type.

DOI: https://doi.org/10.7554/eLife.44964.003

more afferents were labeled in the otolith organs is consistent with more otolith afferents targeting lobe IX than lobe X (***Maklad and Fritzsch, 2003***).

In sum, in the Glt25d2 mouse line the source of Cre+ primary vestibular afferents that project to lobe X are mostly dimorphic afferents in the ipsilateral semicircular canals and extrastriolar regions of the sacculus, and are therefore likely to predominantly convey information about angular acceleration of the head. To investigate all the primary vestibular projections to cerebellum, injections were made using a non-Cre-dependent retrograde virus (AAV2-retro-CAG-GFP) targeting lobe X. In all cases both lobes IX and X were infected (as well as cerebellar nuclei) (***Figure 2—figure supplement 2***). VG ipsilateral to the site of injection had many retrogradely labeled somata, including 2.7% that were calretinin-positive (34/1486, n = 3 VG in separate experiments). The majority of calretinin positive cells were not retrolabeled (91%, 343/377). Examples of central/striolar pure-calyx afferents that were retrolabeled were found in all five end organs, although they were rare, numbering only a few per end organ (***Figure 2—figure supplement 2E***). This provides evidence that some pure-calyx afferents may project to cerebellum, but we cannot determine whether they project to lobe X, lobe IX or cerebellar nuclei, as all regions were infected. In comparison to the Cre-dependent virus, this viral injection labeled many more afferents in all the end organs, but especially in the otolith organs (***Figure 2—figure supplement 2G–H***). In this experiment many peripheral afferents in the lateral utricle were labeled, consistent with the report that hair cell polarity relates to afferent projection pattern, with afferents innervating lateral utricle projecting to cerebellum and medial utricle projecting to vestibular nuclei (***Maklad et al., 2010***). The majority of the afferents appeared to be dimorphic and were too dense/numerous to count accurately. These sources of primary afferent projections to mouse cerebellum were similar to those reported in gerbils (***Purcell and Perachio, 2001***).

## Primary vestibular afferents innervate ON UBCs in cerebellum

Having established that most of the primary afferents to lobe X are dimorphic VG fibers from the semicircular canals, we asked whether these fibers contact UBCs. The Glt25d2 mouse line was crossed with a channelrhodopsin (ChR2) reporter line (Ai32), which caused expression of ChR2 and EYFP in primary vestibular afferents. This cross allowed specific activation of primary afferents with light in acute brain slice physiology experiments. Whole-cell patch-clamp recordings were made near ChR2-EYFP-expressing mossy fiber endings in sagittal slices of cerebellum containing lobe X, specifically targeting recordings to candidate UBC somata identified by size (~10 µm diameter; ***Figure 3A***).

ChR2 activation of primary afferents with blue light flashes caused bursts of action potentials in postsynaptic UBCs (***Figure 3B***). Activation of primary afferents led to time-locked, depressing EPSCs, followed by a slow inward current that began at the end of the stimulation train; both responses were mediated by AMPA receptors, and are diagnostic of ON UBCs (***Figure 3C***) (***Borges-Merjane and Trussell, 2015***; ***Lu et al., 2017***; ***Zampini et al., 2016***). The chances of finding a UBC that happened to be contacted by a nearby labeled fiber was low. However, of 107 UBCs recorded in brain slices from 22 mice, all 13 UBCs that responded to optogenetic activation of primary afferents were ON UBCs. The response to ChR2 stimulation of primary afferents was remarkably similar to responses evoked by electrical stimulation of white matter (***Figure 3D***). Thus, primary afferents preferentially target ON UBCs and we found no evidence for primary projections to OFF UBCs.

Recorded cells were filled with biocytin for post hoc imaging during whole-cell recording (***Figure 3E–F***). Biocytin fills confirmed the UBC morphology of the recorded cells and allowed visualization of the contacts between presynaptic EYFP-labeled mossy fiber axon and biocytin filled postsynaptic brush in six experiments (***Figure 3—figure supplement 1***). This approach provided views of the complex morphology of these synaptic interfaces. 3D renderings were made in order to estimate

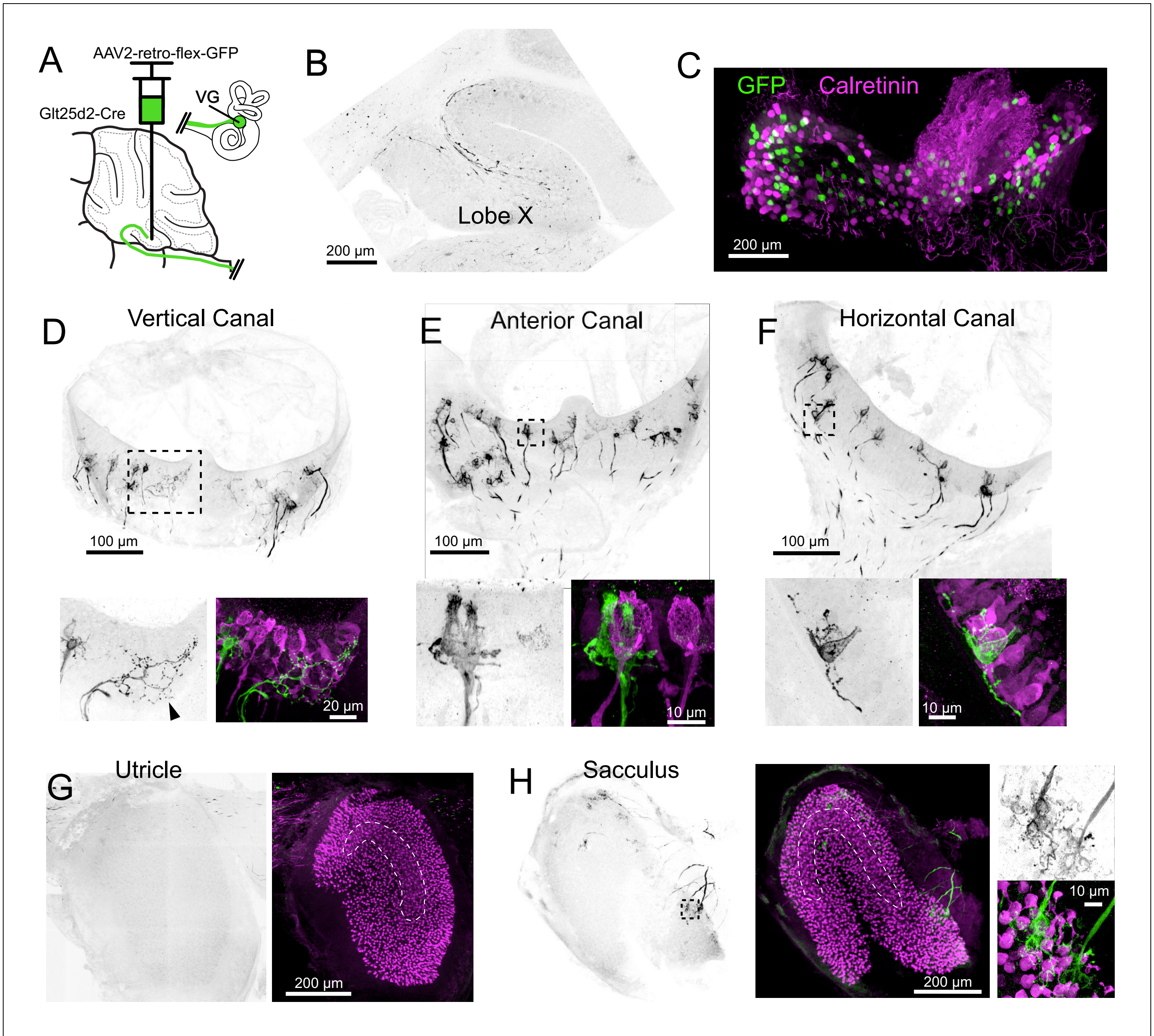


**Figure 2.** Cre+ primary vestibular afferents carry angular acceleration signals from the semicircular canals. (**A**) Cre-dependent retro-AAV was injected into lobe X of Glt25d2 mice to label the primary afferents with GFP to identify their dendritic endings in the vestibular organs. In this figure all of the images are from the same animal. (**B**) At the injection site labeled axons are apparent. GFP- black. (**C**) Somata in the VG were infected and expressed GFP. None colocalized with calretinin, a marker for vestibular primary afferents with pure-calyx endings. (**D**) Retrolabeled dimorphic calyx endings in the crista of the vertical canal. GFP- black. Below- Boxed region above expanded- Examples of retrolabeled (left- black, right- green) dimorphic calyces, a single retrolabeled bouton-only arbor (arrowhead) and calretinin+ pure calyces (magenta). Note that some hair cells also express calretinin in mice. (**E**) Retrolabeled dimorphic calyx endings in the crista of the anterior canal. Below- Boxed region above expanded- note the clearly dimorphic calyces having a flask shape with narrow top. Calretinin-expressing pure-calyx afferents (magenta) have a wider top and no bouton endings. The central region of the cristae only rarely had retrolabeled afferents and they were never pure-calyx. (**F**) Retrolabeled dimorphic calyx endings in the crista of the horizontal canal. Below- Boxed region above expanded. (**G**) The peripheral area of the utricle had few labeled afferents. Calretinin (magenta) was used to determine the extent of the utricular macula and to identify the striola (dashed outline). (**H**) Retrolabeled dimorphic calyces were found in the sacculus having many calyces and boutons per afferent fiber. See also ***Figure 2—figure supplements 1–2***.

DOI: https://doi.org/10.7554/eLife.44964.004

The following figure supplements are available for figure 2:

*Figure 2 continued*

**Figure supplement 1.** Injection that infected lobe IX in addition to lobe X labeled more fibers, especially in the medial utricle.
DOI: https://doi.org/10.7554/eLife.44964.005
**Figure supplement 2.** Non-Cre dependent retro-AAV reveals source of primary vestibular projections to cerebellum.
DOI: https://doi.org/10.7554/eLife.44964.006

the surface area of the brush and the area of the brush that contacted the mossy fiber (***Figure 3G–H***, ***Figure 3—figure supplement 2***). Although this is not a direct measure of the transmitter release regions, it quantifies the area of apparent contact where transmission occurs. The area of the brush that contacted the mossy fiber was 99.45 ± 40.95 $\mu m^2$ (mean ± SD). The area of the UBC brush itself was 446.68 ± 86.35 $\mu m^2$ (mean ± SD), and thus nearly a quarter of the dendrite was available for synaptic contact. We tested whether the morphology of these connections correlated with the synaptic responses of the UBCs. The postsynaptic fast EPSC was positively correlated with UBC brush area, but not the contact area between the mossy fiber and brush (***Figure 3I***). The slow EPSC amplitudes that occur at the offset of stimulation and decay times did not correlate with the contact area between the mossy fiber and UBC or the brush area (***Figure 3J***). This lack of correlation may suggest that the postsynaptic AMPARs that mediate this slow current are at some distance from the sites of contact with the mossy fiber, or that glutamate removal by diffusion or transport shape this current (***Lu et al., 2017***).

## mGluR1-expressing ON UBCs receive Cre+ primary vestibular afferent input while calretinin-expressing OFF UBCs do not

mGluR1 is expressed by ON UBCs and not by OFF UBCs, while calretinin is expressed by OFF UBCs and not by ON UBCs (***Borges-Merjane and Trussell, 2015***). Calretinin expression thus marks pure-calyx afferents of the vestibular end organs, as well as cerebellar OFF UBCs. Immunohistochemical labeling of these two markers of UBC subtype in cerebellar sections of Glt25d2::tdTomato mice expressing tdTomato in primary afferents revealed numerous projections to mGluR1-expressing UBCs, but not to calretinin-expressing UBCs (***Figure 3K–M***), confirming the physiological analysis. To quantify the proportion of UBCs that receive input from these primary vestibular afferents a systematic random sampling approach was taken that ensured all of the granule cell layer of lobe X had an equal probability of being sampled (see Materials and methods). Overall 145 mGluR1-expressing UBCs were counted, 29 of which received primary afferent input (20%). In the same brain sections 96 calretinin+ UBCs were counted, none of which received primary afferent input. Thus, a direct VG projection to lobe X targets mGluR1-expressing ON UBCs but not calretinin-expressing OFF UBCs.

Although the expression of Cre appeared random in the VG (see above), it is possible that the Glt25d2 line may express Cre specifically in a subpopulation of VG neurons that target mGluR1-expressing UBCs, rather than a representative population. To label VG neurons that do not express Cre in the Glt25d2 line, we injected GFP or tdTomato-expressing viruses with different serotypes (AAV9, AAV2-retro, AAV-PHP.S) in the posterior semicircular canal (***Figure 4***). This viral approach infected populations of VG neurons of various sizes presumed to be a different population than those that express Cre in the Glt25d2 mouse line and was therefore a complimentary approach to label diverse VG neuron types (***Figure 4B–E***). Of the VG neurons infected, ~8.4% expressed calretinin and were therefore the 'pure-calyx' type (Out of 636 virus labeled neurons in 4 VG, 54 expressed calretinin). Vestibular primary afferents were labeled in lobe X and their apparent mossy fiber swellings were imaged along with immunohistochemically localized mGluR1 and calretinin-expressing UBCs (***Figure 4F–G***). Of the 240 mossy fiber terminals imaged, 79 mGluR1-expressing UBCs had brushes interdigitated with the terminals (n = 4 mice). No calretinin-expressing UBCs were seen making contact to primary vestibular afferents. The morphology of the primary afferents was striking. They seldom branched as they projected along the white matter in the sagittal plane. The morphology of these primary afferents is compared directly with secondary afferents below. Taken together, these data provide multiple lines of evidence showing that primary vestibular afferents project exclusively to mGluR1-expressing ON UBCs and not to calretinin-expressing OFF UBCs.

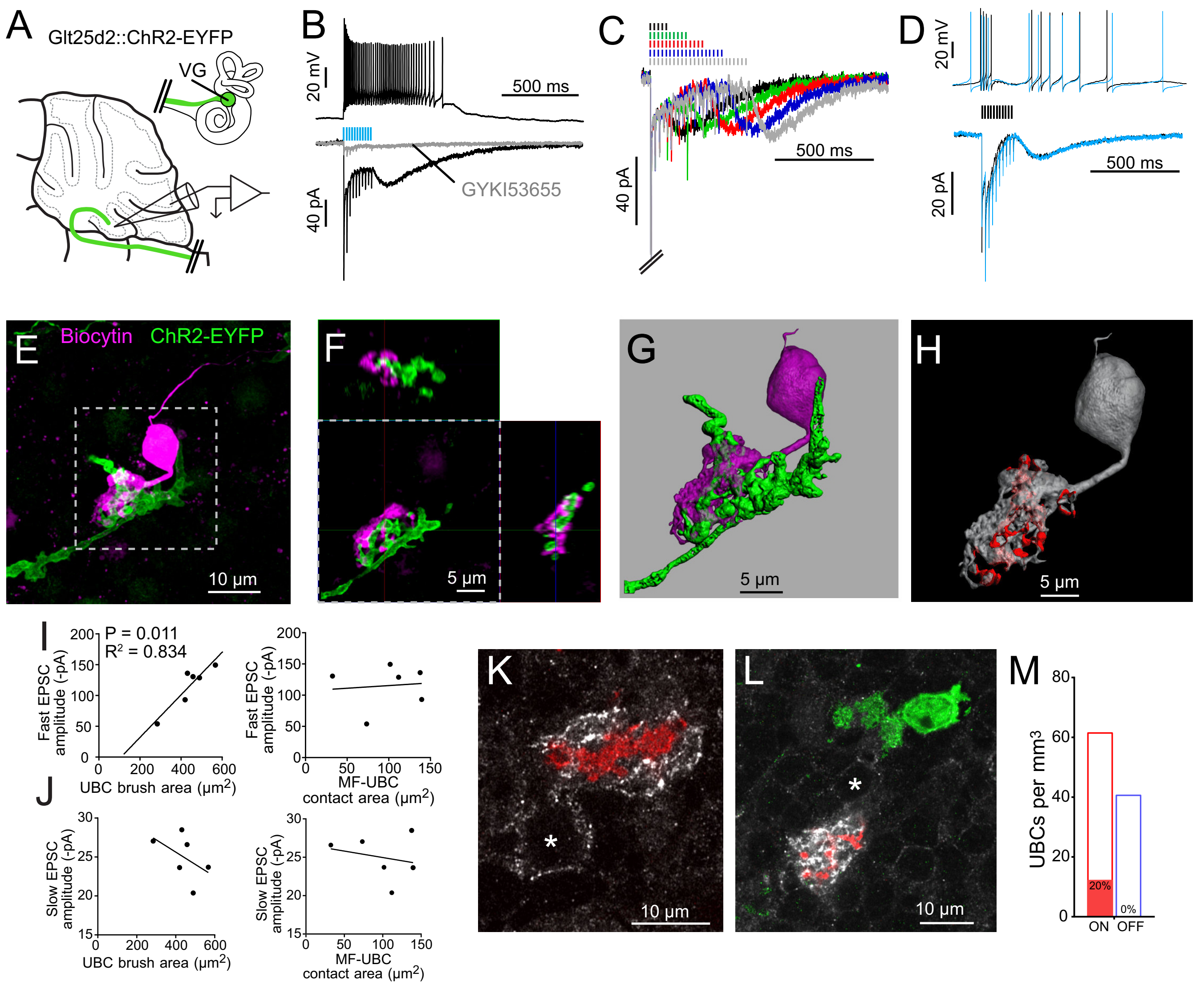


**Figure 3.** Primary vestibular afferents project to ON UBCs in lobe X. (**A**) In the Glt25d2::ChR2-EYFP cross, primary afferents from the vestibular ganglion (VG) expressed ChR2 and were activated by blue light during whole-cell recordings of UBCs in an acute slice preparation. (**B**) Mossy fibers were activated by LED light pulses (50 Hz 0.25 ms) that evoked spiking responses in postsynaptic UBC that outlasted mossy fiber activation. In the same cell in voltage clamp (below) light pulses evoked fast EPSCs that depressed, followed by a slow inward current. 50 µM GYKI53655 blocked the majority of the inward current. This slow AMPAR-mediated current after the offset of stimulation is diagnostic of an ON UBC. This case was without 4-AP in the bath. All UBCs that had light evoked PSCs were ON UBCs (n = 13). (**C**) 50 Hz light stimulation of various train durations illustrated as lines above the traces. The slow AMPAR-mediated current begins at the offset of stimulation, consistent with re-activation of AMPARs as they recover from desensitization while glutamate gradually leaves the synapse. (**D**) Spiking response (top) and EPSCs (bottom) evoked by electrical stimulation (3.8 V, 50 Hz, 0.25 ms, black) were similar to those evoked by ChR2 stimulation (50 Hz 0.25 ms, blue) in the same cell. (**E**) UBCs were filled with biocytin and recovered in 6/13 cases. This UBC received input from ChR2-EYFP expressing primary vestibular afferent. Maximum intensity projection. (**F**) Orthogonal view of the boxed region in B, showing UBC brush wrapping around mossy fiber. (**G**) Surfaces were created on the fluorescence to characterize the structure of the mossy fiber-UBC synapse. (**H**) A one voxel thick contact layer between the UBC and mossy fiber surfaces was made to calculate the apposition area between the two surfaces (shown in red). The calculated apposition area of this mossy fiber to UBC contact was 137.66 $\mu m^2$. (**I**) The postsynaptic EPSC correlated with the area of the UBC brush (left), but did not correlate with the contact area between the mossy fiber and brush (right). Currents are in the presence of 4-AP. (**J**) The slow EPSC did not correlate with the UBC brush area (left) or contact area between the mossy fiber and brush (right), suggesting that this current is due to the action of glutamate at distant receptors. Currents are in the presence of 4-AP. (**K–L**) In Glt25d2::tdTomato cross, tdTomato+ primary afferents were seen innervating the brushes of mGluR1+ UBCs (white), but not calretinin+ UBCs (green). Soma of mGluR1+ UBCs identified with *. Single image planes. (**M**) 20% of counted mGluR1+ UBCs were contacted by tdTomato+ primary afferents. No counted calretinin+ UBCs were contacted by these primary afferents. See also ***Figure 3—figure supplements 1–5***.

*Figure 3 continued*

DOI: https://doi.org/10.7554/eLife.44964.007

The following figure supplements are available for figure 3:

**Figure supplement 1.** Additional UBCs that were filled with biocytin and recovered with an innervating primary afferent expressing ChR2-EYFP.

DOI: https://doi.org/10.7554/eLife.44964.008

**Figure supplement 2.** Imaging, surface construction and measurement of fluorescent spheres of known size.

DOI: https://doi.org/10.7554/eLife.44964.009

**Figure supplement 3.** UBCs and granule cells receive disynaptic inhibitory input, likely via direct Golgi cell activation by primary vestibular afferents.

DOI: https://doi.org/10.7554/eLife.44964.010

**Figure supplement 4.** Granule cells receive primary vestibular synaptic input.

DOI: https://doi.org/10.7554/eLife.44964.011

**Figure supplement 5.** 4-AP has effects on the ChR2-evoked EPSC, but does not change whether the UBC response is ON or OFF type.

DOI: https://doi.org/10.7554/eLife.44964.012

## Primary vestibular afferent pathway evokes disynaptic IPSCs in UBCs and granule cells

Since primary vestibular afferents excite ON UBCs, we asked whether they might also trigger inhibitory control of the same UBCs. Indeed, some UBCs that received direct primary afferents had ChR2-evoked fast disynaptic inhibitory post synaptic currents (IPSCs) in addition to monosynaptic EPSCs. In many cases, activation of primary afferents evoked IPSCs alone in UBCs and granule cells, without a typical ON or OFF synaptic response (***Figure 3—figure supplement 3***). The onset of these IPSCs occurred at a delay consistent with disynaptic inhibition in all cases (5.74 ± 1.42 ms (mean ± SD), n = 23). In most of these UBCs some component of the IPSC was blocked by $GABA_AR$ antagonist SR-95531 and the remaining current was blocked by glycine receptor antagonist strychnine (***Figure 3—figure supplement 3B***). Presumably cerebellar Golgi cells, which co-release GABA and glycine, are the source of this disynaptic inhibition (***Rousseau et al., 2012***). Thus, the same population of primary vestibular afferents both excite ON UBCs and activate a pathway that leads to their inhibition.

## Primary vestibular afferents innervate granule cells in cerebellum

The primary afferents that contacted ON UBCs also contacted granule cells, but generated clearly different physiological responses. Optogenetic activation of Cre+ afferents resulted in fast EPSCs in granule cells, but never exhibited a slow AMPAR-mediated EPSC at the offset of stimulation (***Figure 3—figure supplement 4A***). In contrast, the peak and decay time of the first EPSC in the response train was similar to postsynaptic responses of ON UBCs (***Figure 3—figure supplement 4B***, ***Figure 3—figure supplement 5***). The contact area between mossy fiber and granule cell claw was only about 15% of those measured between primary afferent and UBC brush (***Figure 3—figure supplement 4C–K***). These results are consistent with the hypothesis that the slow EPSC of UBCs results from pre and postsynaptic structure, and is not simply a feature of mossy fiber transmitter release per se. Granule cells also received inhibition at a latency consistent with disynaptic input from Golgi cells (5.52 ± 0.40 ms (mean ± SD), n = 10, ***Figure 3—figure supplement 3C***). These data indicate that Cre+ primary afferents contact granule cells and Golgi cells, but specifically target the ON subtype of UBCs.

## Secondary vestibular afferents innervate ON and OFF UBCs in cerebellum

A major target of the vestibular primary afferents is the medial vestibular nucleus of the brainstem (MV). The principal neurons of MV project secondary vestibular afferents to lobe X, and therefore represent a second potential source of mossy fiber input to UBCs. To target ChR2 to this secondary vestibular pathway, viral injections were made into MV of mGluR2-GFP mice, which express GFP in UBCs (***Borges-Merjane and Trussell, 2015***). The virus (AAV1-CAG-ChR2(H134R)-mCherry) expressed the same variant of ChR2 as that expressed in the Ai32 (ChR2-EYFP) mouse line, fused to mCherry (red fluorescent protein). Three weeks after infection, acute brain slices were prepared and whole-cell patch-clamp recordings of UBCs were made near mCherry-labeled secondary afferents

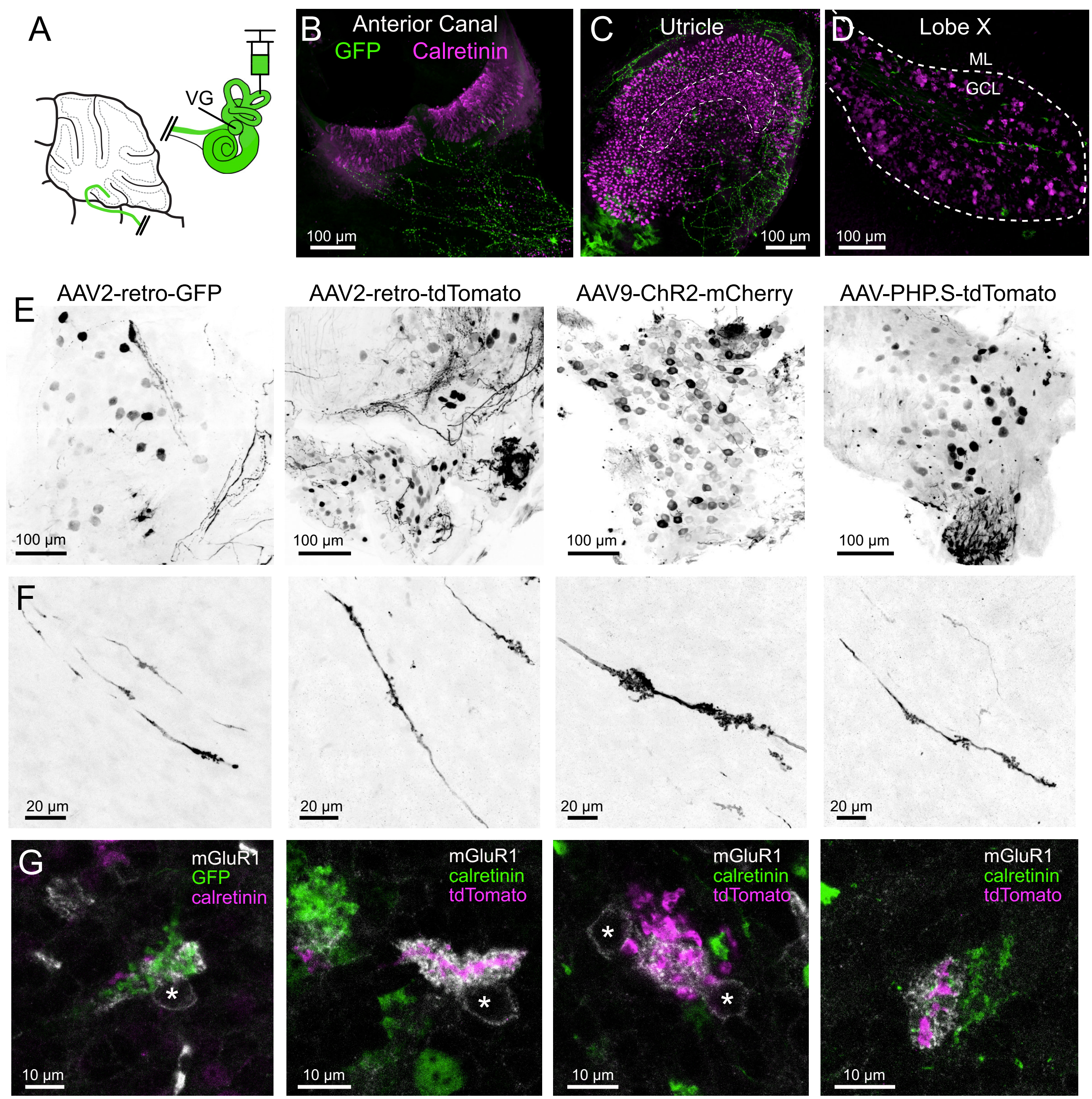


**Figure 4.** AAV infected vestibular primary afferents contact mGluR1-expressing, but not calretinin-expressing UBCs. (**A**) AAVs of various serotypes were injected into the posterior vertical semicircular canal of adult mice. (**B–C**) Example showing experiment using AAV2-retro-CAG-GFP. The anterior canal and utricle had numerous peripheral afferent fibers labeled (green). Calretinin (magenta) labels pure-calyx afferents and Type I hair cells. (**D**) Each 50 µm thick section of lobe X of cerebellum had a few virally-labeled afferents (green) shown among many calretinin-expressing UBCs (magenta). (**E**) Vestibular ganglia showing viral expression of fluorescent proteins (black) using four different viruses in separate experiments indicated above. Many neurons of various sizes were labeled. Images in F-G correspond to ganglia and viruses indicated in E. (**F**) Virally-labeled primary afferents were apparent in lobe X. Note the thickness of the afferents and the lack of branching. (**G**) Immunostaining for calretinin and mGluR1 was used to investigate to which type of UBC this population of fibers projected. Each terminal swelling along virally-labeled primary afferents was imaged, along with calretinin and mGluR1 labeling. 33% of afferent endings intercalated with the brush of an mGluR1+ UBCs, whereas no calretinin+ UBCs were contacted. Somata of mGluR1 + UBCs indicated with *. Single image planes.

*Figure 4 continued*

DOI: https://doi.org/10.7554/eLife.44964.013

(***Figure 5A***). The location of the injection site was histologically confirmed in all experiments. Primary afferent axons local to the injection site were only rarely infected, as identical injections into Glt25d2::ChR2-EYFP mice showed few co-labeled neurons (see below).

Out of 108 UBCs recorded in brain slices from 14 mice, nine postsynaptic UBCs were of the OFF subtype: ChR2 activation of secondary afferents caused initial fast EPSCs that then depressed and led to a slow IPSC that caused a pause in spiking (***Figure 5B***). The IPSC was blocked by the mGluR2 antagonist LY341495 in all cases tested (***Figure 5C***). In four additional cases, secondary vestibular afferents projected to ON UBCs, based on the presence of a slow inward current response (***Figure 5D***). Seven of the OFF UBCs were filled with biocytin and 3D rendered in order to estimate the brush area and contact area between the mossy fiber and the brush (***Figure 5E–H***, ***Figure 5—figure supplement 1A***). As was seen with ON UBCs that received primary input, the fast EPSC amplitude correlated with the UBC brush area of these OFF UBCs (***Figure 5I***). No correlations between the slow IPSC and mossy fiber-UBC contact area or UBC brush area were found (***Figure 5J***). The EPSCs of secondary afferent-receiving ON UBCs were larger than those of secondary afferent-receiving OFF UBCs (***Figure 5K***). EPSCs of ON UBCs that received secondary afferents were similar to ON UBCs that received primary afferents (secondary: 45.17 ± 8.26 pA vs primary: 46.35 ± 14.09 pA, (mean ± sem), t-test, $p=0.954$, $n = 11$). All four ON UBCs were recovered for histological analysis (***Figure 5—figure supplement 1B***). One of the ON UBCs had two brushes, which is a rare morphology (***Braak and Braak, 1993***; ***Mugnaini and Floris, 1994***).

To corroborate these physiological results we took an anatomical approach using the same ChR2-mCherry expression in MV and utilized mGluR1 and calretinin expression to identify ON and OFF UBCs (***Figure 5L***). Of 231 mGluR1+ UBCs counted, 44 (19%) received labeled mossy fiber input. Of 114 calretinin+ UBCs counted, 19 (17%) received labeled mossy fiber input. Thus, although their populations differ in number, a similar proportion of mGluR1+ and calretinin+ UBCs are innervated by secondary afferents.

## Primary and secondary vestibular afferents differ in morphology in cerebellum

Primary and secondary afferents in the cerebellum appeared to have different morphologies (***Figure 6***), suggesting that mossy fiber structure may differ depending upon their source. To compare the primary and secondary afferents in the same sections, a mCherry expressing virus (AAV1-CAG-ChR2(H134R)-mCherry) was injected into the right MV of Glt25d2::ChR2-EYFP reporter mice. ChR2, being a transmembrane protein, targeted the fused mCherry or EYFP proteins to the membranes of primary and secondary afferents. Labeled secondary afferents were more numerous than primary afferents (***Figure 6A–B***), although their number is somewhat artificial given the incomplete labeling of both VG and MV neurons. In addition to lobe X, primary and secondary afferents projected to ventral leaflet of lobe IX, where UBCs are also present in high density relative to other lobes (***Harris et al., 1993***). Primary afferents only projected into IXc, whereas secondary afferents also projected into the more caudal lobe IXb (***Figure 6C–D***). The terminals of primary fibers were often 'rosette-like', similar to those of secondary afferents, but in many cases the elaborate protrusions from the main fiber ran along a longer length of the axon than the more spherically shaped secondary afferents (***Figure 6E–F***).

The thickness of the primary and secondary afferent axons between terminals was clearly different (***Figure 6E–F***). Measurement of axon diameter between rosettes (mean of several measurements along axon >5 µm from mossy terminal swellings) indicated that primary afferents were significantly thicker than secondary afferents (***Figure 6G***). This was also the case for afferents that contacted UBCs that were recovered along with biocytin cell fills in physiology experiments, providing further evidence that primary afferents were not infected by viral injections into MV (***Figure 6H***). In addition, primary afferents labeled by viral injection into the posterior semicircular canal were similar in thickness and morphology to the primary afferents in the Glt25d2::ChR2-EYFP line, and larger than the secondary afferents (Glt25d2-labeled primary afferents: 1.20 ± 0.34 µm, $n = 48$; AAV-labeled primary

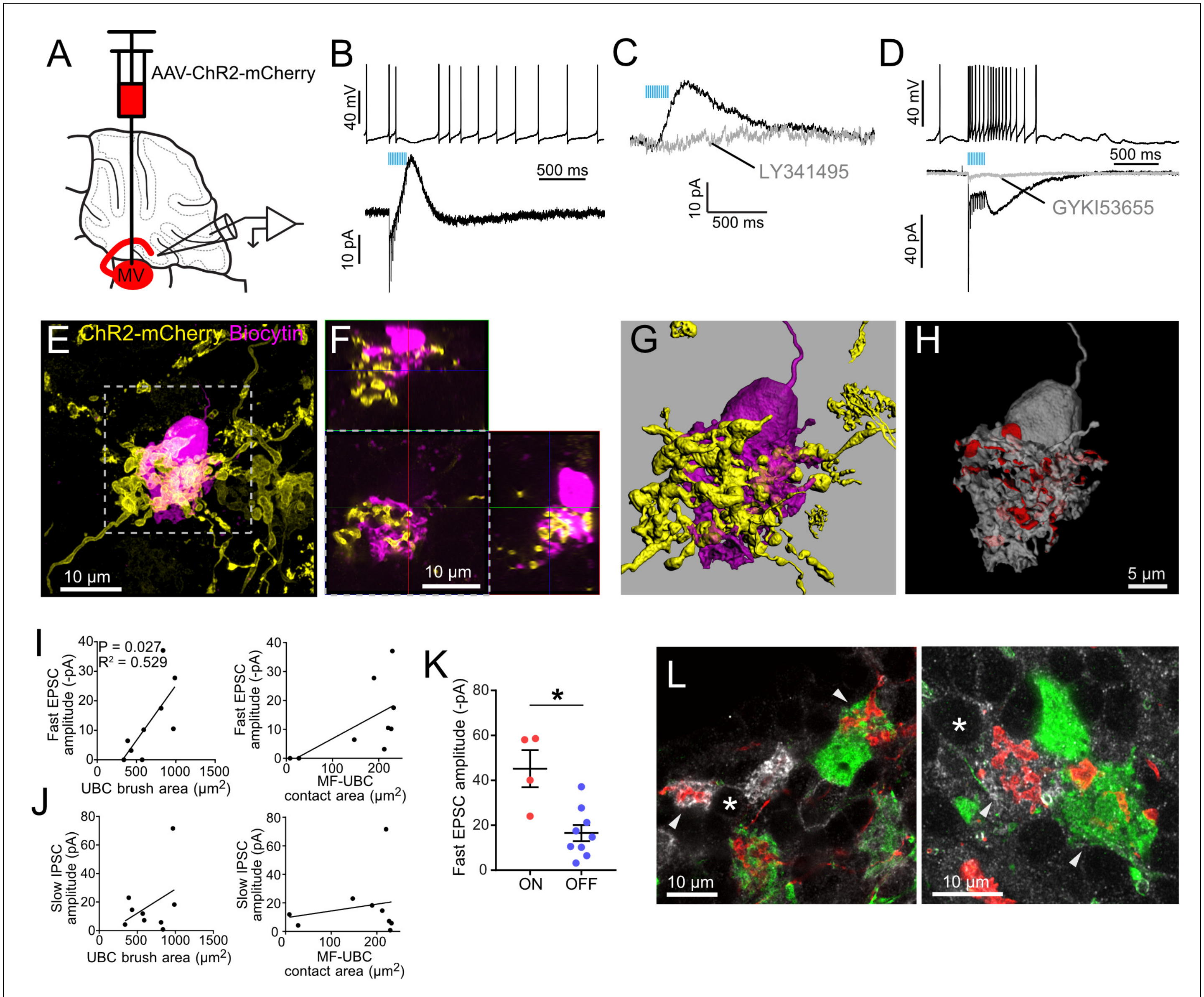


**Figure 5.** Secondary vestibular afferents project to OFF UBCs and ON UBCs. (**A**) Medial vestibular nucleus (MV) was infected with a ChR2 expressing virus. (**B**) Example OFF UBC. 50 Hz 10x ChR2 activation of secondary vestibular mossy fibers caused a pause in spiking (above) and evoked a slow IPSC diagnostic of OFF UBCs (below). (**C**) The AMPAR-mediated fast EPSC varied across cells. In this example it was very small. In OFF UBCs, the IPSC was always blocked by mGluR2 antagonist LY341495 (1µM). (**D**) Example ON UBC. 50Hz10x ChR2 activation of secondary vestibular mossy fibers caused an increase in spiking (above) and evoked a fast and slow EPSC that were blocked by GYKI53655 (50µM) diagnostic of an ON UBCs (below). (**E**) The UBC shown in B was filled with biocytin (magenta) and recovered. Yellow: Immunohistochemical amplification of mCherry which is fused with the ChR2 protein. Maximum intensity projection. (**F**) Orthogonal view of boxed region in E, showing UBC brush wrapping around mossy fiber. (**G–H**) The mossy fiber and UBC shown in E-F was 3D rendered and the apposition area between the two volumizations was calculated. The calculated apposition area of this mossy fiber to UBC contact was 336 $\mu m^2$ (red). (**I**) The postsynaptic EPSC of the OFF UBCs correlated with the area of the UBC brush. $R^2 = 0.529$, p=0.027 (left), but did not correlate with the contact area between the mossy fiber and brush (right). (**J**) The slow IPSC of the OFF UBCs did not correlate with the UBC brush area (left) or contact area between the mossy fiber and brush (right). (**K**) OFF UBCs that received input from secondary afferents had smaller EPSCs than those of ON UBCs. t-test, p=0.003. (**L**) mCherry-expressing secondary mossy fibers (red) were seen innervating the brushes of mGluR1+ UBCs (white) and also calretinin+ UBCs (green). Soma of mGluR1+ UBCs identified with *. Arrowheads indicate contact between mossy fiber and brush. Single image planes. See also ***Figure 5—figure supplement 1***.

DOI: https://doi.org/10.7554/eLife.44964.014

The following figure supplement is available for figure 5:

**Figure supplement 1.** Additional UBCs that responded to secondary vestibular afferent stimulation.

Figure 5 continued

DOI: https://doi.org/10.7554/eLife.44964.015

afferents: 1.13 ± 0.34 µm, n = 58; AAV-labeled secondary afferents: 0.80 ± 0.28 µm, n = 100 (mean ± SD) ***Figure 6G***). Thus the differences between primary and secondary afferents are not due to pathology caused by life-long expression of ChR2-EYFP in the Glt25d2::ChR2-EYFP line.

Surprisingly, the differences in morphology based on source of input also extended to the postsynaptic cells. The UBCs that received secondary afferent input had larger dendritic brushes than UBCs that received primary input. In some cases, the primary-receiving UBC had a brush that wrapped around the primary afferent itself, rather than around an apparent swelling or rosette (***Figure 7A***, ***Figure 3—figure supplement 1***), indicating that release sites can be located at these regions of the axon. Secondary-receiving UBCs were more likely to contact a spherically shaped rosette (***Figure 7B***, ***Figure 5—figure supplement 1***). Measurements of the contact area between afferent and UBC, indicated that the area between primary afferents and UBCs was smaller than the contact area between secondary afferents and UBCs, likely due to the more complex rosettes made by secondary afferents (***Figure 7C***). Even the dendritic brush (including non-synaptic membrane) of secondary-receiving UBCs was larger in area and volume than the brushes of primary-receiving UBCs (***Figure 7D***). These differences were not due to OFF UBCs being larger than ON UBCs, because the secondary-afferent receiving ON UBCs were similar in size to the secondary-receiving OFF UBCs. The contact area between the brush and mossy fiber relative to the entire surface area of the brush was similar between primary and secondary-receiving UBCs (t-test, p=0.180, n = 17). This suggests that the postsynaptic brush develops in such a way to match the anatomy of the earlier maturing mossy fiber (***Ashwell and Zhang, 1998***; ***Sekerková et al., 2004***). Finally, UBCs targeted by primary and secondary afferents even differed in soma size, regardless of ON/OFF subtype (***Figure 7E***). Thus, the global morphology of UBCs is tuned to the source of mossy fiber.

## Primary vestibular afferents generate build-up EPSCs via non-mossy contact to soma

Besides the ON/OFF distinction described previously, some UBCs respond to electrical stimulation of white matter with a peculiar slow-rising EPSC (***Figure 8A–B***) (***Zampini et al., 2016***). These AMPAR-mediated EPSCs are distinct from typical synaptic responses due to their slow activation during the stimulus and slow decay upon cessation of stimulation and their lack of fast EPSCs; notably, they lacked the slow inward current that appears only after transmission ceases, characteristic of the ON UBC. Previously these build-up responses were considered to arise from variation in apposition of receptors and release sites at mossy fiber terminals (***Zampini et al., 2016***). Here we asked if they represent a different form of input with unique origin. Build-up EPSCs were always blocked by AMPAR antagonists which in some cases revealed a small mGluR2-mediated IPSC (***Figure 8B***). In other cases, primary afferent stimulation evoked a small IPSC mediated by mGluR2, which, when blocked revealed the build-up EPSC (***Figure 8D***). Electrical stimulation of white matter activates all axons nearby, including primary, secondary and intrinsic mossy fibers from UBCs. Therefore, the source and mechanism underlying these build-up EPSCs are not easily studied using conventional approaches. In the present experiments that utilized Glt25d2::ChR2-EYFP mice to stimulate primary afferents selectively, access to pre- and postsynaptic morphology allowed us to investigate the basis of these build-up responses in detail.

ChR2-evoked build-up EPSCs were small (***Figure 8A***), with a 6.06 ± 2.05 pA (mean ± SD) peak in response to ten stimuli at 50 Hz, and had a decay time constant of 445.38 ± 302.43 ms (n = 5 UBCs). EPSCs in response to single light flashes could be resolved and were smaller than responses to trains. In all four cases in which build-up EPSCs were evoked by primary afferent optogenetic activation and the UBC was recovered, there was no mossy fiber contact to the UBC brush (***Figure 8C–D***). Rather, in all of these cases, primary afferents contacted UBC somata. Additionally, in all cases of recovered UBCs that did not have apparent build-up EPSCs, none had a ChR2-expressing mossy fiber contacting the soma (n = 5 primary-receiving UBCs) (cf. ***Figure 3***, ***Figure 3—figure supplement 1***). Thus, the build-up response represents activity of an unconventional UBC input, but generated by a primary afferent.

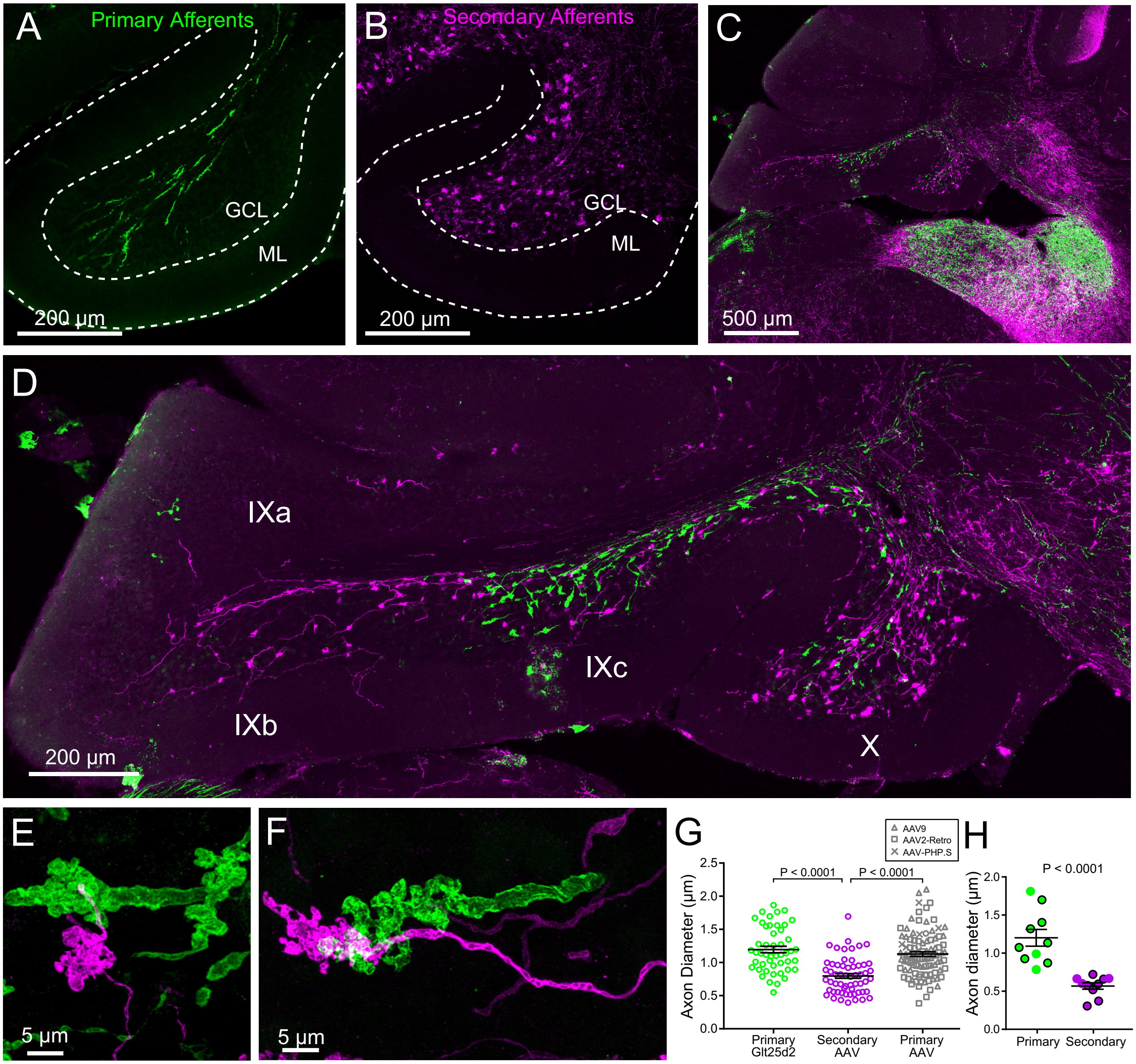


**Figure 6.** Primary and secondary vestibular mossy fiber morphology. (**A**) Expression of a fluorescent protein in primary vestibular afferents (green). Note central location in the granule cell layer (GCL). ML- molecular layer. (**B**) Expression of a fluorescent protein in secondary vestibular mossy fibers (magenta). Note distribution of mossy fibers throughout the width of the GCL. (**C–D**) In a separate set of experiments Glt25d2::ChR2-EYFP mice had AAV1-CAG-ChR2-mCherry injected into MV, which allowed labeling of both primary (green) and secondary (magenta) mossy fiber projections in the same brain sections. Note the intermingling of primary and secondary fibers in the brainstem vestibular nuclei. Primary afferents only projected to rostral lobe IX (IXc), whereas secondary fibers innervated the entire ventral leaflet (IXb and IXc). Careful observation revealed few colabeled mossy fibers, suggesting possible but unlikely confound of virus infected primary fibers. (**E–F**) Higher magnification of primary and secondary vestibular afferents in the same section. Note the larger diameter axon of the primary afferent (green). The apparent overlap of the mossy fibers in these maximum intensity projections is due to the position of one above the other in the z dimension. (**G**) The mean diameter of primary afferents was significantly thicker than that of secondary afferents. Data points are individual mossy fiber axons. ANOVA, $p=0.0001$, post hoc t-tests, $p<0.0001$, $n = 206$ axons. (**H**) The diameter of primary and secondary afferents that contacted UBCs that were recorded from and recovered. Data points with black outline are UBCs that had ChR2 expressing mossy fiber making major contact with brush. Data points without outline had ChR2 expressing mossy fiber making small contact with brush and the ChR2-evoked response was small. t-test, $p<0.0001$, $n = 20$ axons.

DOI: https://doi.org/10.7554/eLife.44964.016

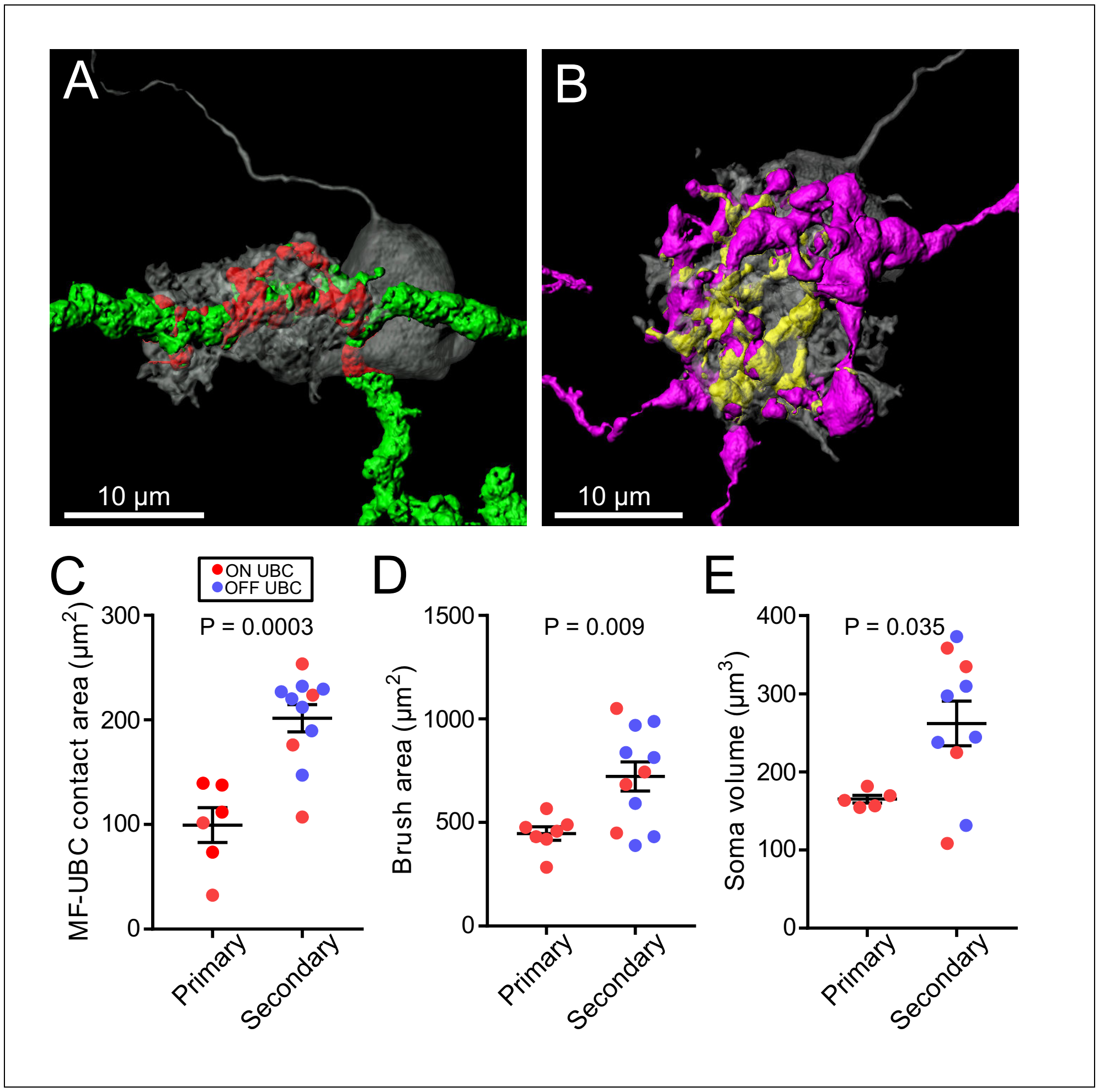


**Figure 7.** UBC dendritic brush size relates to source of input and not ON or OFF UBC subtype. (**A**) 3D rendering of primary afferent showing characteristic thick axon (green). Biocytin fill- gray, contact between mossy fiber and UBC- red. (**B**) 3D rendering of secondary mossy fiber showing thin axons and rosette-like ending. Biocytin fill- gray, contact between mossy and UBC- yellow. (**C**) The contact areas between recovered primary afferents and UBC brushes was smaller than those of secondary mossy fibers (t-test, p=0.0003, n = 17). (**D**) The surface area of the UBC brushes that received primary afferents were smaller than those of secondary mossy fiber-receiving UBCs (t-test, p=0.009, n = 18). The volume of the UBC brushes that received primary afferents were also larger than those of secondary mossy fiber-receiving UBCs (t-test, p=0.014, n = 18), not shown. (**E**) The volume of the somas of UBCs that received primary afferents were smaller than those of secondary mossy fiber-receiving UBCs (t-test, p=0.035, n = 15).

DOI: https://doi.org/10.7554/eLife.44964.017

If the build-up response is due to somatic synapses, then a UBC receiving both contact to the brush and to the soma would be predicted to have a typical ON UBC response (due to the brush contact) with an additional build-up EPSC (due to the soma contact). Indeed, in a primary-receiving ON UBC that had contact with the same mossy fiber on both the brush and the soma, the ChR2-evoked response was a combination of the typical fast EPSC plus a build-up EPSC (***Figure 8E***). Both currents were mediated by AMPARs, as they were blocked by GYKI. Shank1, a postsynaptic density protein, confirmed that postsynaptic receptors may be present at the somatic membrane in regions that appear to contact the primary afferent (***Figure 8E***).

In some UBCs the build-up EPSC could be evoked by either ChR2 stimulation or electrical stimulation (***Figure 8F***). At higher electrical stimulation intensity a generic ON UBC response appeared, but it could not be evoked by ChR2 stimulation. This may have been due to low ChR2 expression in the mossy fiber. In 3 UBCs with the build-up response to ChR2 stimulation, further electrical

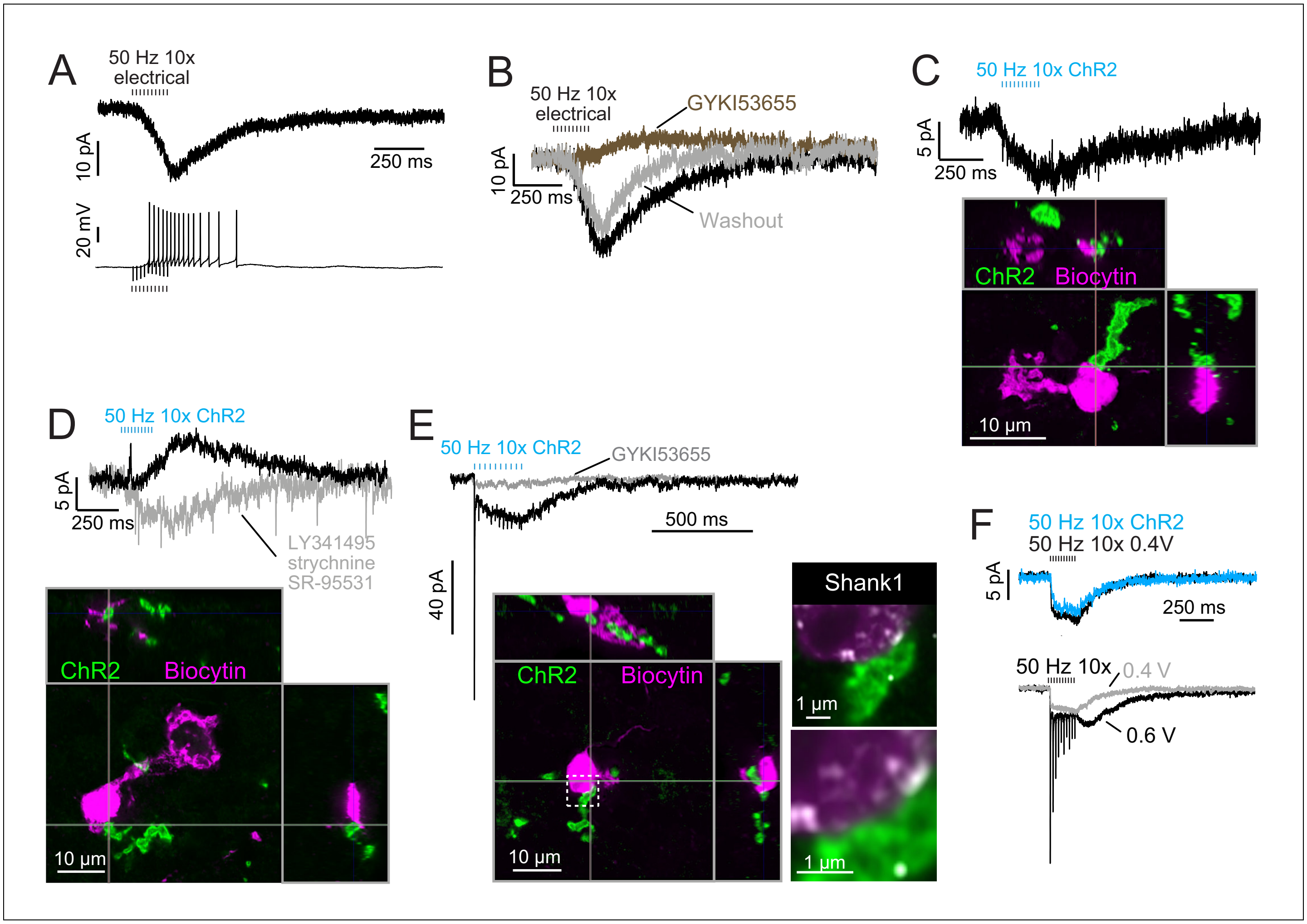


**Figure 8.** Build-up AMPAR-mediated EPSCs can be evoked by ChR2 or electrical evoked transmitter release and may be due to somatic synapses. (**A**) Slow build-up EPSC evoked by presynaptic electrical stimulation in UBC. The build-up of inward current is sufficient to cause a burst of spikes beginning at the 5th stimulus. No 4-AP in bath. These currents are distinct from the usual slow EPSC seen in ON UBCs that occurs at the stimulus offset. (**B**) Another UBC with a similar slow EPSC evoked by 50 Hz train of presynaptic electrical stimulation. The current was blocked by AMPAR antagonist GYKI53655 (50 µM), revealing a small IPSC presumably mediated by mGluR2 known to be present in UBCs. (**C**) Slow EPSC evoked by ChR2 stimulation of primary vestibular afferent. This UBC was filled with biocytin and recovered post hoc. Super-resolution imaging revealed the mossy fiber made contact with the soma of the UBC and not the brush. (**D**) IPSCs evoked by ChR2 stimulation of primary vestibular afferent. The fast transient IPSC was blocked by strychnine (0.5 µM) and SR-95531 (5 µM) and the delayed slower IPSC was blocked by LY341495 (1 µM). Biocytin cell fill revealed that this UBC received input to the soma and to the shaft of its dendritic brush. This mGluR2 mediated IPSC was seen in 4 UBCs that received primary afferent input. The slow IPSC amplitude for primary afferent receiving UBCs was 5.43 ± 2.89 pA (mean ± SD), n = 4. (**E**) This UBC had a response that appeared to be a combination of dendritic and somatic synapses due to the fast EPSC at stimulus onset and buildup of inward current during the train, respectively. Below- Recovery of the cell and the input mossy fiber revealed that this was indeed possible, because the primary vestibular afferent made contact with both the soma and the dendritic brush. This is the cell shown in ***Figure 7A***. Right- Boxed region expanded- Shank1 antibody staining (white) was seen on the periphery of the soma directly opposed to the primary afferent. (**F**) Slow EPSC evoked by 50 Hz train of either low intensity electrical synaptic stimulation (0.4 V, black) or ChR2 stimulation (blue). The similarity between responses suggests that the same input is being stimulated and the transmitter release is comparable between stimulation methods. Below- the same UBC with two levels of electrical synaptic stimulation. The observation that a stronger electrical stimulation evokes a typical ON UBC response suggests that this cell may have received multiple inputs.

DOI: https://doi.org/10.7554/eLife.44964.018

stimulation evoked an ON response. In no case did electrical stimulation evoke an OFF UBC response, and thus it may be that axosomatic synapses are only made onto ON UBCs.

## Discussion

The vestibular cerebellum is unique in its high density of UBCs, suggesting a cerebellar processing function of UBCs specific to the vestibular system. UBCs fall into two classes, ON and OFF, based on their response to mossy fiber input. This study demonstrates that different cerebellar input pathways differentially recruit these response classes based on extraordinary specificity of innervation (**Figure 9**). Although ON and OFF cells are co-distributed in lobe X, primary afferents project to ON UBCs exclusively. These primary afferents specifically are the fibers of VG neurons whose peripheral endings are largely in the semicircular canals, and make both calyx and bouton endings (dimorphic VG fibers). Secondary afferents from MV contact OFF UBCs as well as ON UBCs. UBCs targeted by the primary vs secondary afferents also differed in synaptic and dendritic morphology, and even soma size. Finally, a build-up form of excitatory UBC response appeared to be due to synaptic contact between the mossy fiber and the postsynaptic soma. Thus, the specificity of vestibular projection includes not just the originating vestibular organ, but subtypes of VG and UBC cells, with corresponding anatomical and physiological refinements.

### UBCs as an input layer preceding granule cells

Purkinje cells are considered the site of multimodal integration in the cerebellum, due to their enormous number of granule cell inputs. Recent studies have highlighted the integrative aspects of cells in the granule cell layer as well. Granule cells have multiple dendrites, allowing them to receive signals from multiple modalities (**Chabrol et al., 2015**; **Huang et al., 2013**; **Knogler et al., 2017**; **Sawtell, 2010**). By contrast, UBCs receive only a single mossy fiber input to their dendritic brush and therefore do not integrate multiple modalities, instead maintaining the activities of ensembles of postsynaptic granule cells segregated in a 'labeled line'. Such an arrangement may be of particular advantage in cerebellar vestibular processing vs other cerebellar modalities. The typical pattern of integration by granule cells could disrupt vestibular processing by mixing inputs from the five vestibular end organs (per ear) that sense head movements in different directions. Instead, UBCs could act

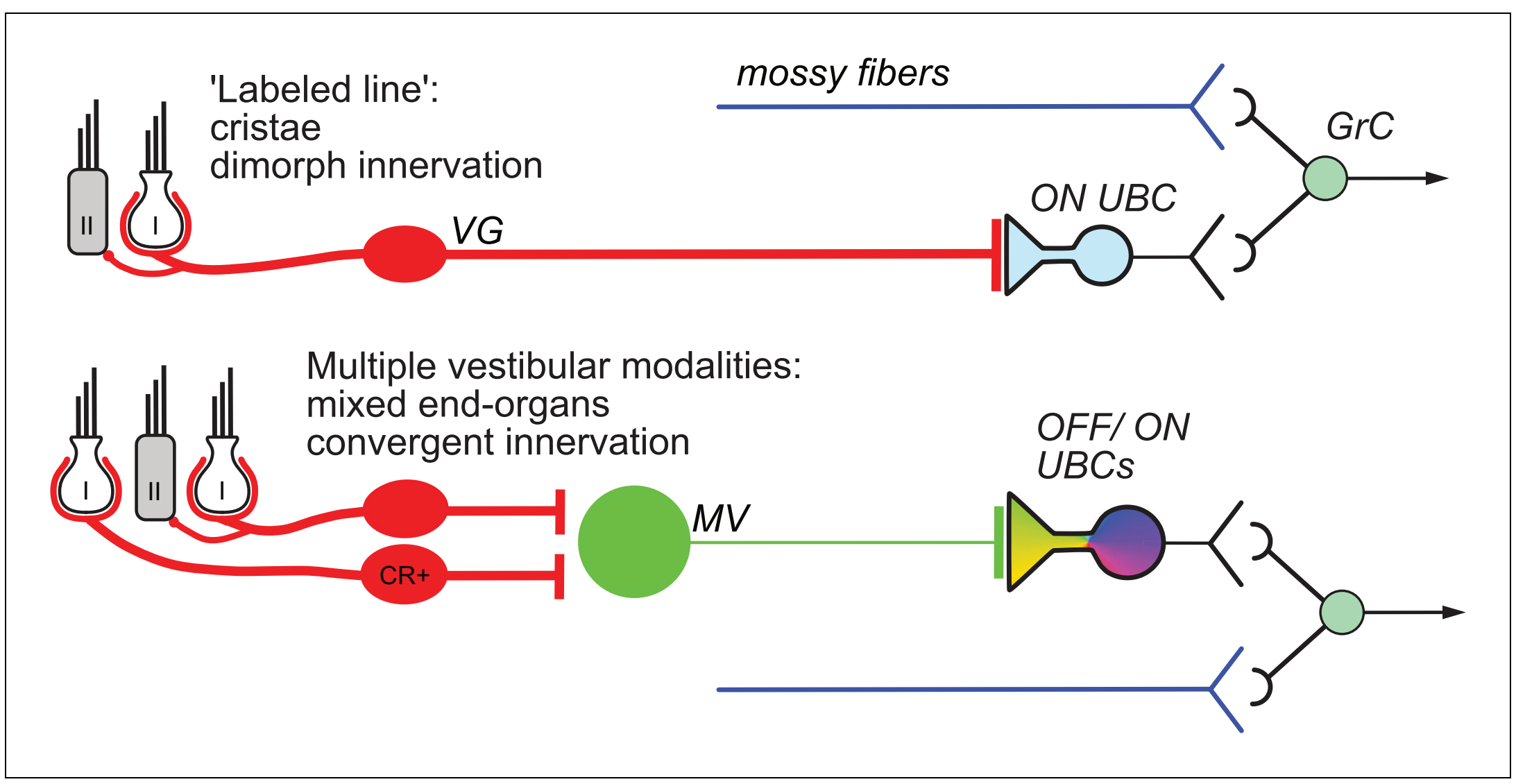


**Figure 9.** Summary of results. Vestibular ganglion (VG) neurons having dimorphic peripheral endings that receive input from Type I and II hair cells project their primary afferents directly to ON UBCs in lobe X of cerebellum, and do not target OFF UBCs. These primary afferents also project to granule cells as well as inhibitory interneurons (Golgi cells) that provide feed-forward inhibition to ON UBCs (not shown). This direct projection to cerebellum may coordinate ensembles of granule cell activities in a 'labeled line' representing acceleration or velocity of the head in a single direction. Secondary mossy fibers arising from neurons in the medial vestibular nucleus (MV) project to both OFF and ON UBCs. MV neurons receive input from calretinin-expressing (CR+) VG neurons that make pure-calyx dendritic endings around Type I hair cells, in addition to dimorphic primary afferents. The signals carried by secondary afferents are integrated across vestibular end organs and processed by the local MV circuit. Thus, distinct anatomical pathways that vary in the convergence of inputs, level of processing and axon morphology target distinct UBC classes to transform specific sensory signals.

DOI: https://doi.org/10.7554/eLife.44964.019

as an input layer prior to the granule cells to allow divergence to parallel ensembles of postsynaptic granule cells that each faithfully represent head movement along the axes of different end organs. Convergence must occur at some point to integrate signals from the canals and otoliths, which is necessary to estimate orientation relative to gravity, and this may happen at the granule cell and/or Purkinje cell level. Alternatively, or in addition, convergence occurring in the MV could be processed specifically by UBCs that receive secondary input. Further experiments are necessary to explore whether primary and secondary pathways target distinct populations of granule cells, either through UBCs or directly, and whether the primary and secondary-receiving granule cells vary in physiological response or morphology, as do UBCs.

## Secondary mossy fibers from MV relay integrated signals to UBCs

MV neurons receive multiple primary afferent inputs, feedback inhibition from Purkinje cells, and can be inhibited by stimulation of the contralateral vestibular organs (**Shimazu and Precht, 1966**; **Uchino et al., 1986**). Thus, secondary vestibular mossy fibers may carry signals integrated from multiple end organs and both hemispheres to both ON and OFF UBCs. This is a strikingly different pattern of connectivity than the ON UBCs that receive primary afferent input from a cluster of hair cells in a single end organ and a single VG subtype. OFF UBCs do not appear to receive input from primary afferents at all, and may therefore only process signals that have been integrated by MV. This circuitry indicates that OFF UBCs process bilateral vestibular signals to pause input to ensembles of granule cells, perhaps in a push-pull circuit that could contribute to reflexive eye movements.

The rate of input to secondary afferent-receiving UBCs may be preserved by MV neurons, which are known to respond to synaptic input with high-fidelity EPSCs whose amplitudes are rate-invariant (**McElvain et al., 2015**). In addition, granule cells respond faithfully to mossy fiber input in vivo, responding to a burst of mossy fiber input with a burst of action potentials that is similar in duration (**Arenz et al., 2008**; **Chadderton et al., 2004**). It is at UBCs where profound signal processing occurs through highly rate-dependent EPSC amplitudes and long duration responses that may even be inverted by OFF UBCs (**Kennedy et al., 2014**). This extended response could be particularly important in the cerebellum. A longer duration burst of action potentials conveyed to the parallel fiber-Purkinje cell synapse will be more likely to drive the Purkinje cell because of facilitation at this synapse and would extend the window of integration within which climbing fiber activity can influence circuit learning.

At the level of the vestibular granule cell, it is likely that a single neuron integrates both primary and secondary inputs (**Chabrol et al., 2015**) but also intrinsic mossy fiber input from UBCs. (**Chabrol et al., 2015**) emphasized that mossy fibers from different sources may exhibit different forms of short-term plasticity, and these characteristic time-dependent responses impact the integrative function of the granule cell. Given the radical transformation of mossy fiber input by UBCs, which results in prolonged or delayed firing, or cessation in activity in vivo (**Kennedy et al., 2014**), granule cells that receive some dendritic input from a UBC's intrinsic mossy fiber likely will be dominated by that input while the UBC is active. However, paired recordings between presynaptic UBCs and postsynaptic granule cells will be necessary to test this hypothesis. Additionally, whether a single granule cell integrates input from multiple UBCs carrying primary and secondary signals will be an important next step to understanding the integration that occurs in vestibular cerebellum.

## Primary and secondary vestibular afferents have distinct morphologies in cerebellum

**Brodal and Drablos (1963)** suggested that vestibular lobes of rat cerebellum contain a population of mossy fibers that differ from those of other lobes. The fact that they reported these fibers in flocculus, despite a dearth of primary afferents (**Newlands and Perachio, 2003**; **Osanai et al., 1999**), implies that these fibers may have been intrinsic mossy fibers of UBCs, as suggested by **Rossi et al. (1995)**. Differences in morphology between mossy fibers based on their source have been reported in non-vestibular lobes of the cerebellum. Mossy fibers projecting from the deep cerebellar nuclei are larger, are more likely to have filipodia projecting from the rosette, and have more boutons, than those projecting from basal pontine nuclei (**Gao et al., 2016**; **Gilmer and Person, 2017**). While mossy fibers originating from different regions that project to lobe X vary in presynaptic plasticity (**Chabrol et al., 2015**), we find that these axons also exhibit characteristic morphological features

that may support differences in electrical activity level. Primary afferents were quite thick, projected along the white matter of lobe X, and only rarely branched. It is perhaps not surprising that the primary vestibular afferents in the cerebellum are large given that their diameter in the vestibular nerve is among the thickest in the brain (mean, ~3 µm) (**Gacek and Rasmussen, 1961**); notably, these also have elevated tonic firing rates of >100 Hz (**Jones et al., 2008**). The relatively thin diameter of the secondary afferents suggests lower firing rates (**Perge et al., 2012**). Indeed, vestibular nucleus neurons that respond to vestibular stimulation in vivo have spontaneous firing rates between 0 and 30 Hz in cats and ~65 Hz in squirrel monkeys (**Cullen and McCrea, 1993**; **Shimazu and Precht, 1965**). In mice, the best approximation of spontaneous firing of secondary mossy fibers may be the EPSCs recorded in granule cells in the flocculus that could be modulated by vestibular stimulation. These EPSCs occurred at ~13 Hz under anesthesia (**Arenz et al., 2008**), much lower than vestibular nerve fibers.

Some UBCs that were postsynaptic to primary afferents had their dendritic brush wrapping around smooth parts of the axon, providing anatomical evidence that synapses may exist along the length of the axon in addition to the terminal swellings. This is corroborated by the finding that smooth parts of the primary afferent contacting a UBC soma could evoke EPSCs. Such differences in *en passant* mossy terminal morphology might affect efficiency of propagating action potentials.

## Atypical synaptic input to UBCs

In build-up responses, contacts are made directly to the UBC soma, clearly out of reach of the dendrite. Previous descriptions of build-up responses in UBCs speculated that such responses might arise from misalignment of a mossy fiber active zone relative to an AMPAR cluster (**Zampini et al., 2016**). Instead, the observation that build-up EPSCs occur specifically when mossy fibers appear to contact UBC somas demonstrates a novel basis for these synaptic currents. This conclusion depended upon recovering many filled cells after optogenetically stimulating labeled mossy fibers. Apparently, UBC somata express some AMPA receptors sufficient to respond to somatic inputs. Indeed, outside-out patch-clamp recording has previously shown that AMPARs do function in somatic membranes of UBCs (**Kinney et al., 1997**). Previous analysis of electron micrographs highlighted mossy fiber terminals that contacted Golgi cell somata, forming large convoluted '*en marron*' synapses (**Chan-Palay and Palay, 1971**), which are distinct from the club-like endings contacting UBCs. Mossy fibers touching granule cell somata have also been observed, although these same mossy fibers only made definitive synaptic contacts with nearby granule cell dendrites (**Palay and Chan-Palay, 1974**). The fortuitous observation of somatic contacts by mossy fibers that were associated with distinct postsynaptic responses suggests that somatic inputs could represent a previously unappreciated form of transmission in the granule region of the cerebellum.

# Materials and methods

### Key resources table

| Reagent type (species) or resource | Designation | Source or reference | Identifiers | Additional information |
|---|---|---|---|---|
| Genetic reagent (*M. musculus*) | C57BL/6J | Jackson Laboratory | RRID: IMSR_JAX:000664 | |
| Genetic reagent (*M. musculus*) | B6.Tg(Colgalt2-cre) NF107Gsat/Mmucd | Dr. Chip Gerfen (NIH) PMID: 20023653 | RRID: MGI:2138232 | Referred to as Glt25d2 |
| Genetic reagent (*M. musculus*) | B6.TgN (grm2-IL2RA /GFP)1kyo | Dr. Robert Duvoisin (OHSU) PMID: 9778244 | RBRC: RBRC01194 | |
| Genetic reagent (*M. musculus*) | Ai9(RCL-tdT) | Jackson Laboratory PMID: 22446880 | RRID: IMSR_JAX:007909 | |
| Genetic reagent (*M. musculus*) | Ai32 (RCL-ChR2 (H134R)/EYFP) | Jackson Laboratory PMID: 22446880 | RRID: IMSR_JAX:024109 | |

*Continued*

| Reagent type (species) or resource | Designation | Source or reference | Identifiers | Additional information |
|---|---|---|---|---|
| Genetic reagent (*M. musculus*) | B6.Cg-Et (tTA/mCitrine) TCGOSbn | Dr. Adam Hantman (Janelia Farm) PMID: 26999799 | | |
| Antibody | Chicken polyclonal anti-GFP | Aves Labs | Cat # GFP-1020 RRID: AB_10000240 | IHC (1:2000) |
| Antibody | Rabbit polyclonal anti-DsRed | Clontech | Cat# 632496 RRID:AB_10013483 | IHC (1:2000) |
| Antibody | Goat polyclonal anti-mCherry | Sicgen | Cat# AB0040-200 RRID:AB_2333092 | IHC (1:2000) |
| Antibody | Mouse monoclonal anti-rat mGluR1a | BD Pharmingen | Cat# 556389 RRID:AB_396404 | IHC (1:800) |
| Antibody | Rabbit polyclonal anti-calretinin | Swant | Cat# 7697 RRID:AB_2619710 | IHC (1:500–2000) |
| Antibody | Goat polyclonal anti-calretinin | Swant | Cat# CG1 RRID:AB_10000342 | IHC (1:500–2000) |
| Antibody | Rabbit polyclonal anti-shank1 | Synaptic Systems | Cat# 162 013 RRID:AB_2619859 | IHC (1:1000) |
| Antibody | Mouse monoclonal anti-Myo7A | Dr. Peter Barr-Gillespie (OHSU) | | IHC (1:500) |
| Antibody | Donkey polyclonal anti-chicken Alexa Fluor 488 | Jackson Immuno Research Labs | Cat# 703-545-155 RRID:AB_2340375 | IHC (1:500) |
| Antibody | Donkey polyclonal anti-mouse Alexa Fluor 488 | Jackson Immuno Research Labs | Cat# 715-545-150 RRID:AB_2340846 | IHC (1:500) |
| Antibody | Donkey polyclonal anti-rabbit Cy3 | Jackson Immuno Research Labs | Cat# 711-165-152 RRID:AB_2307443 | IHC (1:500) |
| Antibody | Donkey polyclonal anti-goat Cy3 | Jackson Immuno Research Labs | Cat# 705-165-147 RRID:AB_2307351 | IHC (1:500) |
| Antibody | Donkey polyclonal anti-mouse Alexa Fluor 647 | Jackson Immuno Research Labs | Cat# 715-605-151 RRID:AB_2340863 | IHC (1:500) |
| Antibody | Donkey polyclonal anti-chicken Alexa Fluor 647 | Jackson Immuno Research Labs | Cat# 703-605-155 RRID:AB_2340379 | IHC (1:500) |
| Antibody | Streptavidin-Alexa Fluor 647 | ThermoFisher Scientific | Cat# S21374 RRID:AB_2336066 | IHC (1:2500) |
| Chemical compound, drug | Alexa Fluor 594 hydrazide sodium salt | ThermoFisher Scientific | Cat# A10438 | |
| Chemical compound, drug | GYKI-53655 | Tocris | Cat # 2555 | |
| Chemical compound, drug | JNJ-16259685 | Tocris | Cat # 2333 | |

*Continued*

| Reagent type (species) or resource | Designation | Source or reference | Identifiers | Additional information |
|---|---|---|---|---|
| Chemical compound, drug | LY-341495 | Tocris | Cat # 1209 | |
| Chemical compound, drug | (+)-MK-801 hydrogen maleate | Sigma | Cat # M107 | |
| Chemical compound, drug | Strychnine hydrochloride | Sigma | Cat # S8753 | |
| Chemical compound, drug | SR-95531 hydrobromide | Tocris | Cat # 1262 | |
| Chemical compound, drug | 4-Aminopyradine | Tocris | Cat # 940 | |
| Recombinant DNA reagent | AAV1-CAG-ChR2(H134R) -mCherry (2.92E12 GC/ml) | University of Pennsylvania Vector Core | Cat # CS0949 | |
| Recombinant DNA reagent | AAV9-CAG-ChR2(H134R) -mCherry (2.96E12 GC/ml) | University of Pennsylvania Vector Core | Cat # CS0916 | |
| Recombinant DNA reagent | AAV2-retro-CAG-GFP (1.0E13 GC/ml) | Janeila Farm Vector Core PMID: 27720486 | | Dr. Adam Hantman (Janelia Farm) |
| Recombinant DNA reagent | AAV2-retro-CAG-tdTomato (7.0E12 GC/ml) | Addgene PMID: 27720486 | Cat # 59462-AAVrg | |
| Recombinant DNA reagent | AAV2-retro-CAG-Flex-GFP (9.86E12 GC/ml) | Janeila Farm Vector Core PMID: 27720486 | | Dr. Adam Hantman (Janelia Farm) |
| Recombinant DNA reagent | AAV-PHP.S-CAG-tdTomato (1.7E13 GC/ml) | Addgene PMID: 28671695 | Cat # 59462-PHP.S | |
| Software, algorithm | pClamp 10 | Molecular Devices | RRID:SCR_011323 | |
| Software, algorithm | Igor Pro 6 | WaveMetrics | RRID:SCR_000325 | |
| Software, algorithm | Prism 7 | GraphPad | RRID:SCR_002798 | |
| Software, algorithm | Excel | Microsoft | RRID:SCR_016137 | |
| Software, algorithm | Imaris | Bitplane | RRID:SCR_007370 | |
| Software, algorithm | Zen Black | Zeiss | RRID:SCR_013672 | |
| Software, algorithm | FIJI | https://fiji.sc | RRID:SCR_002285 | |
| Software, algorithm | ImageJ | https://imagej.nih.gov/ij/ | RRID:SCR_003070 | |
| Software, algorithm | Affinity Designer | Serif | RRID:SCR_016952 | |

## Animals

C57BL/6J-TgN(grm2-IL2RA/GFP)1kyo (referred to as mGluR2-GFP) of both sexes were used to identify UBCs (***Borges-Merjane and Trussell, 2015***; ***Nunzi et al., 2002***; ***Watanabe et al., 1998***). Male C57BL/6J-Tg(Colgalt2-cre)NF107Gsat/Mmucd (referred to as Glt25d2) mice were used to express either tdTomato or ChR2-EYFP in primary vestibular afferents by crossing with Ai9(RCL-tdT) (Jackson Labs 007909) (***Madisen et al., 2010***) or Ai32(RCL-ChR2(H134R)/EYFP) (Jackson Labs 024109) (***Madisen et al., 2012***) mouse lines, respectively. The TCGO mouse line was used for its

sparse granule cell labeling (C57BL/6J.Cg-Et(tTA/mCitrine)TCGOSbn) (***Huang et al., 2013***; ***Shima et al., 2016***). Wild type C57BL/6J mice were used for semicircular canal injections and for breeding. Mouse lines were maintained in the animal facility managed by the Department of Comparative Medicine and all procedures were approved by the Oregon Health and Science University's Institutional Animal Care and Use Committee and met the recommendations of the Society for Neuroscience. Because mossy fiber and UBC synapse formation is mature in animals older than postnatal day 21 (P21) (***Morin and Wood, 2001***), we used pups older than this age (P21-P39) for experiments.

## Immunohistochemistry

Mice were overdosed with isoflurane and perfused through the heart with 0.01M phosphate buffered saline, 7.4 pH (PBS) followed by 4% paraformaldehyde in PBS. Brains were extracted from the skull and incubated in the same solution overnight at 4˚C. Brains were transferred to 30% sucrose in PBS for >2 days. 50 µm thick sections were made on a cryostat (HM 550, Microm) at −22˚C and saved as floating sections in PBS. When labeling mGluR1 and calretinin, brains were transferred to PBS instead of 30% sucrose and sectioned on a vibratome. To recover cells that were filled with Biocytin during whole-cell recording, acute brain slices were fixed overnight in 4% paraformaldehyde in PBS, followed by storage in PBS. Both floating 50 µm sections and 300 µm thick acute slices were treated with the following procedures. Sections were rinsed 3 × 10 min in PBS, blocked and permeabilized in 2% BSA, 2% fish gelatin, 0.2% Triton X-100 in PBS for >2 hr at room temperature. Sections were incubated in primary antibodies for 2–3 days at 4˚C on an orbital shaker. Sections were rinsed 3 × 10 min in PBS, followed by secondary antibodies and streptavidin for 2–3 days at 4˚C on an orbital shaker. See Key Resource table for a full list of antibodies used. Sections were rinsed in PBS and in some cases incubated in 4% paraformaldehyde in PBS for 1 hr. Sections were mounted on microscope slides and coverslipped with CFM-3 (CitiFluor).

## Vestibular end organ histology

Mice were perfused with saline with 10 U/ml heparin warmed to 37˚C, followed by 35 ml 4% PFA in 0.1M phosphate buffer 4˚C. End organs were carefully dissected out in PBS and permeabilized and blocked in 2% Triton X-100, 5% normal donkey serum in PBS 1 hr RT shaking. Primary antibodies were incubated for 1–3 days at 4˚C shaking, then rinsed in PBS and incubated in secondary antibodies as above. End organs were coverslipped using a 0.12 mm spacer and CFM-3 mountant.

## Histological imaging and analysis

Images were acquired on a Zeiss Elyra PS.1 with AiryScan system that reconstructs super-resolution images from a series of images acquired under spatially structured illumination (***Gustafsson, 2000***). Images were processed in Zen Black or transferred to Imaris (Bitplane), a multidimentional analysis program based on fluorescence intensity data. Surfaces were created on the channels that contained the UBC and mossy fiber fluorescence to isolate the structure and extract the area and volume statistics and the 3D reconstructions. A surface calculation that is part of the Imaris software was used to create a one voxel thick contact layer between the UBC and mossy fiber surfaces and the contact area was calculated. To test the ability to measure surface areas and volumes accurately, fluorescent microspheres (Spherotech, FP4060-2) were imaged following the same procedures used for biocytin filled cells (***Figure 5—figure supplement 1***).

To count UBCs innervated by primary or secondary mossy fibers, sagittal sections from Glt25d2::tdTomato mice or wild type mice that received MV injections (identical to those made for physiology experiments) were labeled with anti-DsRed, anti-mGluR1a and anti-calretinin as above. Hemispheres contralateral and ipsilateral to the injection were separated by a cut down the midline. Every third 50 µm thick section was histologically labeled. Sections were sampled using a pseudorandom, systematic sampling scheme throughout the mediolateral extent of lobe X. Two fields per section were counted at a random location within the granule cell layer of lobe X. This sampling scheme ensured that every part of the granule cell layer of lobe X had an equal probability of being sampled. Counting was done on a Zeiss LSM 780 confocal microscope using a 63 × 1.4 NA oil immersion objective. A 50 µm x 50 µm unbiased counting frame was used in which UBC somata touching two of the edges were omitted and somata touching the other two edges were included. Calretinin+ or mGluR1+ UBCs were first identified through the depth of the slice. Then whether tdTomato+ mossy

fibers innervated the brush of each UBC was noted. UBCs were only counted when both the brush and the soma were apparent, in order to avoid counting calretinin+ or mGluR1+ mossy fibers that often look similar to UBC brushes. Counts of both UBC types in the same fields ensured the ratios of UBC types would not be affected by their known differential distribution in dorsal vs ventral leaflets of lobe X (**Nunzi et al., 2002**).

In the experiment that used semicircular canal injections to label VG neurons, sections were prepared as above, with anti-GFP, anti-mGluR1a and anti-calretinin. The labeled afferents were more sparse than in Glt25d2, so every labeled terminal swelling was imaged along with mGluR1 and calretinin. Contacts between these virally-labeled primary afferents and mGluR1 or calretinin-expressing UBC brushes were counted in sections throughout lobe X, ipsilateral to the injected inner ear.

To measure primary afferent diameter, sections of lobe X from Glt25d2::ChR2-EYFP amplified with an anti-GFP antibody were imaged. To measure secondary afferent diameter, sections of lobe X from mice injected with AAV1-CAG-ChR2(H134R)-mCherry and amplified with an anti-DsRed or anti-mCherry antibody. Images were captured on a confocal microscope using 63 × 1.4 NA oil immersion objective. Axon diameters > 10 µm from mossy fiber terminal rosettes were measured using ImageJ. 4–8 spans across the axon were measured at ~5 µm intervals and the average was taken as the diameter. Post hoc imaging of axons that projected to biocytin filled UBCs from acute slice experiments were measured in the same way. One secondary afferent that projected to a recorded ON UBC was omitted because its 2.2 µm diameter was >5 SD above the mean diameter (0.8 µm) of secondary afferents and larger than any measured primary afferent. This afferent may be from a cell type present in low number in the MV, but more work is needed to identify the origin of such fibers.

To count peripheral vestibular afferents, whole mounted end organs were imaged using a Zeiss LSM 880 with fast Airyscan super-resolution and 25 × 0.8 NA oil immersion objective. Images were counted using ImageJ and the Cell Counter plugin. Dimorphic calyces counted when they had (1) a 3-dimentional calyx shape (2) at least one bouton process and (3) a labeled axon extending from its base. Calretinin staining clearly labeled pure-calyx afferents that are also distinguishable from their wider opening at the top. These counts are likely underestimates for the total number of retrolabeled afferents, due to tissue damage and inadequate fluorescence. More bouton-only endings may be present because they may be interpreted as being boutons extending from neighboring dimorphs. Afferent fibers were counted 10–50 µm distal to the base of the hair cells.

## Acute brain slice preparation

Mice were anesthetized with isoflurane and decapitated. The brain was rapidly extracted into ice-cold high-sucrose artificial cerebral spinal fluid solution (ACSF) containing (in mM): 87 NaCl, 75 sucrose, 25 $NaHCO_3$, 25 glucose, 2.5 KCl, 1.25 $NaH_2PO_4$, 0.5 $CaCl_2$, 7 $MgCl_2$, bubbled with 5% $CO_2$/95% $O_2$. Parasagittal cerebellum sections containing lobe X were cut at 300 µm with a vibratome (VT1200S, Leica) in ice-cold high-sucrose ACSF. Immediately after cutting, slices were incubated in 35°C recording ACSF for 30–40 min, followed by storage at room temperature. Recording ACSF contained (in mM): 130 NaCl, 2.1 KCl, 1.2 $KH_2PO_4$, 3 Na-HEPES, 10 glucose, 20 $NaHCO_3$, 2 Na-pyruvate, 2 $CaCl_2$, 1 $MgSO_4$, 0.4 Na-ascorbate, bubbled with 5% $CO_2$/95% $O_2$(300–305 mOsm).

## Electrophysiology

Slices were transferred to submerged recording chamber and perfused with the ACSF heated to 33–35°C at 3 ml/min (TC-324B, Warner Instruments). Slices were viewed using an infrared Dodt contrast mask and a 60X water-immersion objective (LUMPlanFL, Olympus) and camera (IR-1000, Dage-MTI) on a fixed stage microscope (Axioskop 2 FS Plus, Zeiss). In slices from mGluR2-GFP mice UBCs were identified by their GFP fluorescence. In slices from Glt25d2 mice UBCs were identified by their soma diameter ~10 µm in the granular cell layer in lobe X. All cells recorded were filled with 1 µM Alexa Fluor 594 hydrazide sodium salt (A10438, Molecular Probes) in order to confirm UBC or granule cell morphology. Pipettes were pulled from thin-walled borosilicate glass capillaries (1.2 mm OD, WPI) to a tip resistance of 5–8 MΩ. The internal pipette solution contained (in mM): 113 K-gluconate, 9 HEPES, 4.5 $MgCl_2$, 0.1 EGTA, 14 Tris-phosphocreatine, 4 $Na_2$-ATP, 0.3 Tris-GFP, with osmolality adjusted to ~290 mOsm with sucrose and pH adjusted to pH 7.3 with KOH. In some experiments 0.1–0.5% biocytin (B1592, Molecular Probes) was added to the pipette solution. Reported voltages are corrected for a −10 mV liquid junction potential. Whole-cell recordings were amplified (10X),

low-pass filtered (10 kHz Bessel, Multiclamp 700B, Molecular Devices) and digitized using pClamp software (20–50 kHz, Digidata 1550, Molecular Devices). Further digital filtering was performed offline, in most cases a 1 kHz low-pass Bessel 8-pole filter was applied. Series resistance was compensated with correction 20–40% and prediction 60–70%, bandwidth 2 kHz. Cells were voltage-clamped at −70 mV. Mossy fibers were stimulated extracellularly by applying voltage pulses (1–50 V, 100–250 µs) using a stimulus generator (Master 8, A.M.P.I.) via a concentric bipolar electrode (CBBPC75, FHC). ChR2 was activated using full-field blue LED light flashes (Lambda TLED+, Sutter) through a GFP filter set.

In some cases, a low concentration (50 µM) of the $K^+$ channel blocker 4-aminopyridine (4-AP) was used to increase the reliability of ChR2-evoked transmitter release, presumably by lowering spike threshold. These cases are indicated in figure legends. Bath application of 4-AP increased the peak EPSC, increased synaptic depression and slowed the decay of the EPSC, but did not change the ON or OFF UBC response type (***Figure 3—figure supplement 5***). Additionally, OFF UBCs were recorded in these slices with electrical stimulation of the white matter and in the presence of 4-AP, indicating that OFF UBCs were present in these transgenic animals and that 4-AP did not block the inwardly rectifying $K^+$ channels that mediate the OFF response (***Figure 3—figure supplement 5***).

## Viral injections

Viral injections were made into the medial vestibular nucleus in P21-25 mGluR2-GFP mice using a stereotax (David Kopf) single axis manipulator (MO-10, Narishige) and pipette vice (Ronal) under isoflurane anesthesia. Glass capillaries (WireTrol II, Drummond Scientific) were pulled on a horizontal puller (P-97, Sutter), beveled at a ~45 degree angle with a 20–30 µm inside diameter using a diamond lapping disc (0.5 µm grit, 3M DLF4XN_5661X) The scalp was cut and a small hole was drilled in the skull. The pipette was lowered into the brain at ~10 µm / s. Five-min periods before and after injection were allowed. 20–50 nl of virus was injected using stereotaxic coordinates 6.1 mm caudal, 0.8 mm lateral to bregma and 3.75 mm ventral to the surface of the brain. AAV1-CAG-ChR2 (H134R)-mCherry (2.92E12 GC/ml) virus from the University of Pennsylvania vector core was injected into MV to label and express ChR2 in secondary mossy fibers. AAV2-retro-CAG-Flex-GFP (9.86E12 GC/ml) or AAV2-retro-CAG-GFP (1.0E13 GC/ml) (Janelia Farm) was injected into lobe X of adult Glt25d2 mice (>12 weeks) using stereotaxic coordinates 7.2 mm caudal, 0.5 mm lateral to bregma and 3.0 mm ventral to the surface of the brain. 200–400 nl of virus was used. Experiments were done 2–3 weeks after virus injection.

Semicircular canal injection was done following ***Suzuki et al. (2017)*** using AAV2-retro-CAG-GFP (1.0E13 GC/ml, Janelia Farm), AAV2-retro-CAG-tdTomato (7.0E12 GC/ml, Addgene) or AAV-PHP.S-CAG-tdTomato (1.7E13, Addgene) (***Chan et al., 2017***), AAV9-CAG-ChR2(H134R)-mCherry (2.96E12 GC/ml, University of Pennsylvania) under isoflurane anesthesia. Briefly, a small hole was bored into the posterior semicircular canal using a 27 ga needle. After a 5-min period to allow the fluid leakage to slow, an injection pipette fused to PE10 tubing followed by polyimide tubing (0.0039' ID, 0.0049' OD) was inserted into the hole and secured in place with muscle and tissue adhesive (Vetbond). 2 µl volume of the virus was injected at 100 nl/min. After 5 min, the tube was removed, the hole was plugged with muscle and sealed with tissue adhesive. Mice were perfused two weeks later.

# Acknowledgements

We would like to thank Sacha Nelson and Adam Hantman for the TCGO mice, Adam Hantman and Kim Ritola for the retro-AAVs, Jocelyn Krey for help with inner ear dissection, Ruby Larisch and Jennifer Goldsmith for help with mouse husbandry, Peter Barr-Gillespie for the Myo7A antibody, Aurelie Snyder, Stephanie Kaech Petrie and Crystal Chaw for help with microscopy, and NIH F32 DC014878, NIH K99 DC016905, and Hearing Health Foundation Emerging Research Grant to TSB, NIH R01 NS028901 and DC004450 to LOT, NIH P30 NS0618000 to S Aicher.

# Additional information


## Funding

| Funder | Grant reference number | Author |
|---|---|---|
| National Institutes of Health | F32 DC014878 | Timothy S Balmer |
| Hearing Health Foundation | Emerging Research Grant | Timothy S Balmer |
| National Institutes of Health | K99 DC016905 | Timothy S Balmer |
| National Institutes of Health | R01NS028901 | Laurence O Trussell |
| National Institutes of Health | DC004450 | Laurence O Trussell |

The funders had no role in study design, data collection and interpretation, or the decision to submit the work for publication.


## Author contributions

Timothy S Balmer, Conceptualization, Data curation, Software, Formal analysis, Funding acquisition, Validation, Investigation, Visualization, Methodology, Writing—original draft, Writing—review and editing; Laurence O Trussell, Conceptualization, Data curation, Supervision, Funding acquisition, Validation, Methodology, Writing—original draft, Project administration, Writing—review and editing


## Author ORCIDs

Timothy S Balmer http://orcid.org/0000-0002-8864-5465
Laurence O Trussell https://orcid.org/0000-0003-1171-2356


## Ethics

Animal experimentation: All procedures were approved by the Oregon Health and Science University's Institutional Animal Care and Use Committee (IP00000952) and met the recommendations of the Society for Neuroscience.


## Decision letter and Author response

Decision letter https://doi.org/10.7554/eLife.44964.022
Author response https://doi.org/10.7554/eLife.44964.023


# Additional files

## Supplementary files

- Transparent reporting form

DOI: https://doi.org/10.7554/eLife.44964.020

## Data availability

All data generated or analysed during this study are included in the manuscript and supporting files.

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
