## [Decision Letter]

Thank you for submitting your article "Selective targeting of unipolar brush cell subtypes by cerebellar mossy fibers" for consideration by *eLife*. Your article has been reviewed by three peer reviewers, including Vatsala Thirumalai as the Reviewing Editor and Reviewer #1, and the evaluation has been overseen by and Eve Marder as the Senior Editor. The following individual involved in review of your submission has also agreed to reveal their identity: Fabrice Ango.

The reviewers have discussed the reviews with one another and the Reviewing Editor has drafted this decision to help you prepare a revised submission.

Summary:

In this manuscript, the authors demonstrate remarkable specificity in connections from vestibular afferents to cerebellar uniploar brush cells (UBCs), representing parallel information streams. Using genetics, tract tracing, super-resolution microscopy and patch clamp recordings, they show that primary afferents from the vestibular end organs target ON UBCs while secondary projections from the vestibular nucleus contact both ON and OFF UBCs. Thus, their data suggest that distinct circuits are used to convey modality-specific information. The manuscript is written clearly and the figures are beautiful! Overall, this is an impressive piece of work and, it contributes largely to our understanding of the vestibulocerebellar circuit organization.

Essential revisions:

A few improvements could be made, detailed below:

1) The authors use a Cre line in several experiments, Glt25d2-Cre that could be better characterized for the reader. The authors note that Cre-driven reporter expression is present in the somata of vestibular ganglia but not vestibular nuclei, but no mention is made of whether Cre expression is seen elsewhere in the brainstem or spinal cord where it could form other sources of mossy fibers. The experiments in which AAV-ChR2 is expressed directly in the VG mitigate some concerns associated with whether the main result of selective UBC innervation will hold, but it is still important for the authors to address more comprehensively the expression pattern of GLt25d2 if the crossed ChR2-reporter mice are used for physiological analysis to assess whether any other precerebellar structures possess somatic label. (i.e. Figure 3).

2) Figure 1: a better characterization of pure bouton single-end organs with anti-peripherin antibody can be performed to discriminate between dimorphic and pure bouton endings (A. Lysakowski et al., 1999). The authors do not detect pure Calyx ending with Glt25d2-Cre line using AAV virus but in Figure 1 Glt25d2::tdTomato colocalized with calretinin in VG. How do the authors interpret this discrepanciy? -The authors suggest somatic synapses on UBC, will it be possible to show post-synaptic receptors labeling on mGluR1 + UBC soma to strengthen the hypothesis?

3) The authors argue that primary afferent input to ON UBCs conveys angular acceleration information based on their innervation of the semi-circular canal hair cells and not the otoliths. But this seems to be true only in the Glt25d2 Cre line and not when retrograde viral tracers were injected into lobe X (Figure 2—figure supplement 2). In this case, several afferents were labeled in the otoliths also. We suggest that the authors address this point in the manuscript or soften the statement made.

---

## [Author Response]

Essential revisions:A few improvements could be made, detailed below:1) The authors use a Cre line in several experiments, Glt25d2-Cre that could be better characterized for the reader. The authors note that Cre-driven reporter expression is present in the somata of vestibular ganglia but not vestibular nuclei, but no mention is made of whether Cre expression is seen elsewhere in the brainstem or spinal cord where it could form other sources of mossy fibers. The experiments in which AAV-ChR2 is expressed directly in the VG mitigate some concerns associated with whether the main result of selective UBC innervation will hold, but it is still important for the authors to address more comprehensively the expression pattern of GLt25d2 if the crossed ChR2-reporter mice are used for physiological analysis to assess whether any other precerebellar structures possess somatic label. (i.e. Figure 3).

The other source of mossy fiber projections to lobe X, in addition to vestibular ganglion and vestibular nuclei, is nucleus prepositus hypoglossi (NpH), which carries eye movement information (Barmack, 2003). The figure has been modified to include additional micrographs showing vestibular nuclei and NpH more clearly (Figure 1C).

2) Figure 1: a better characterization of pure bouton single-end organs with anti-peripherin antibody can be performed to discriminate between dimorphic and pure bouton endings (A. Lysakowski et al., 1999). The authors do not detect pure Calyx ending with Glt25d2-Cre line using AAV virus but in Figure 1 Glt25d2::tdTomato colocalized with calretinin in VG. How do the authors interpret this discrepanciy? -The authors suggest somatic synapses on UBC, will it be possible to show post-synaptic receptors labeling on mGluR1 + UBC soma to strengthen the hypothesis?

We chose not to stain for peripherin to identify bouton afferents because they were quite easy to identify in the end organs based on their morphology alone. Moreover, the low number of bouton-only afferents would make an experiment such as this difficult due to the low probability of infecting them with a virus. We agree that identifying the projections of peripherin expressing VG neurons is of interest. The experiment we would like to do and would be more likely to work would be characterizing the projections in a peripherin-cre mouse line, which have been made, but are not easily available. Thus, this experiment is unfortunately beyond the scope of the current project.

We interpret the lack of pure-calyx endings retrolabeled in the Glt25d2 line as a lack of projections of these afferents to lobe X. This agrees with several observations made in this paper. Central regions of the end organs where the pure-calyx endings are numerous did not appear to project to lobe X. The primary afferents in cerebellum labeled in the Glt25d2::tdTomato reporter cross were not immunoreactive for calretinin. Not shown in the manuscript is the observation that in a calretinin-cre (Calb2tm2.1(cre/ERT2)Zjh) crossed to a reporter no primary afferents are apparent in the cerebellum, see Author response image 1.

Many granule cells and UBCs are labeled and the UBCs mossy fibers can be seen in some cases. By comparing Author response image 1 with Figure 1B, we suggest that the primary afferents are not labeled with this calretinin-reporter approach. This mouse line approach is complimentary to immunostaining, because the cells that express calretinin will be labeled throughout, whether or not calretinin protein is present in the afferent.

Shank1, a postsynaptic density protein that has been used to label postsynaptic areas in granule cells (Huang et al., 2013), was labeled in some experiments. Many puncta are labeled in locations expected along the brush of UBCs and granule cell dendrites. In agreement with the possibility of somatic synapses, puncta are also seen on the surface of UBC somata. This staining has been added to Figure 8E to show an example of a UBC that has these proteins on the soma at the interface with a primary afferent that appeared to elicit a somatic current.

3) The authors argue that primary afferent input to ON UBCs conveys angular acceleration information based on their innervation of the semi-circular canal hair cells and not the otoliths. But this seems to be true only in the Glt25d2 Cre line and not when retrograde viral tracers were injected into lobe X (Figure 2—figure supplement 2). In this case, several afferents were labeled in the otoliths also. We suggest that the authors address this point in the manuscript or soften the statement made.

The non-cre-dependent virus did infect more otolith afferents, especially lateral utricle. However, we cannot conclude that the labeled afferents are from lobe X, as this experiment infected lobe IX and deep cerebellar nuclei as well. Indeed, there is evidence that lateral utricle afferents project to lobe IX, that medial utricle afferents project to vestibular nuclei, and that neither project many afferents to lobe X (Maklad and Fritzsch, 2003; Maklad et al., 2010). This suggests the otolith afferents labeled may be from lobe IX. This point has been clarified in the manuscript.